# Membrane depolarization mediates both the inhibition of neural activity and cell-type-differences in response to high-frequency stimulation
Jae-Ik Lee [1] ✉, Paul Werginz[1,2], Tatiana Kameneva [3,4], Maesoon Im[5,6,7] & Shelley I. Fried [1,8]

Neuromodulation using high frequency (>1 kHz) electric stimulation (HFS) enables preferential activation or inhibition of individual neural types, offering the possibility of more effective treatments across a broad spectrum of neurological diseases. To improve effectiveness, it is important to better understand the mechanisms governing activation and inhibition with HFS so that selectivity can be optimized. In this study, we measure the membrane potential ($V_m$) and spiking responses of ON and OFF α-sustained retinal ganglion cells (RGCs) to a wide range of stimulus frequencies (100–2500 Hz) and amplitudes (10–100 μA). Our findings indicate that HFS induces shifts in $V_m$, with both the strength and polarity of the shifts dependent on the stimulus conditions. Spiking responses in each cell directly correlate with the shifts in $V_m$, where strong depolarization leads to spiking suppression. Comparisons between the two cell types reveal that ON cells are more depolarized by a given amplitude of HFS than OFF cells—this sensitivity difference enables the selective targeting. Computational modeling indicates that ion-channel dynamics largely account for the shifts in $V_m$, suggesting that a better understanding of the differences in ion-channel properties across cell types may improve the selectivity and ultimately, enhance HFS-based neurostimulation strategies.

High-frequency electric stimulation (HFS) enables innovative forms of neuromodulation to be realized, e.g., blocking neural activity[1–4] or preferentially activating (or inhibiting) specific types of fibers[5–8] or neurons[9,10]. Clinical studies that harness some of the innovative features of HFS have demonstrated promising results, including sustained pain reduction in patients with postamputation[11] or chronic back/lower limb pain[12], as well as weight loss in obese individuals[13]. Unlike pharmacological agents, the effectiveness of HFS is almost instantaneous[4,14–16], with no evidence of deterioration of residual function[11], and it has not been associated with any considerable adverse effects even after chronic use[17,18].

Intriguingly, individual types of nerves or nerve fibers within a heterogeneous population can exhibit different sensitivities to HFS, e.g., nerve fibers that carry pain signals can be blocked while fibers that carry other

sensations or mediate motor activity are not disrupted[19,20]. Such selectivity is especially attractive for applications in which a large diversity of neural types are present, e.g., in the retina, upwards of 15 different types of ganglion cells (RGCs, retinal output neurons)[21,22] each elicit distinct patterns of spiking in response to a given light stimulus[23–25]. It is challenging, however, to create distinct patterns of spiking in adjacent neurons with a prosthesis and the transmission of non-physiological signals to the brain likely contributes to the sub-optimal results obtained to date. The ability to selectively target individual types with a prosthesis offers the potential to reproduce important elements of natural signaling and therefore improve clinical outcomes. As one example, ON and OFF types of retinal ganglion cells (RGCs) have similar sensitivity to individual biphasic pulses[26,27] and thus the out-of-phase spiking that occurs naturally in these two types is difficult to replicate with

[1]Department of Neurosurgery, Massachusetts General Hospital, Harvard Medical School, Boston, MA, USA. [2]Institute of Biomedical Electronics, TU Wien, Vienna, Austria. [3]School of Science, Computing, and Engineering Technologies, Swinburne University of Technology, Hawthorn, VIC, Australia. [4]Department of Biomedical Engineering, University of Melbourne, Parkville, VIC, Australia. [5]Brain Science Institute, Korea Institute of Science and Technology (KIST), Seoul, South Korea. [6]Division of Bio-Medical Science & Technology, KIST School, University of Science and Technology (UST), Seoul, South Korea. [7]KHU-KIST Department of Converging Science and Technology, Kyung Hee University, Seoul, Republic of Korea. [8]Boston VA Healthcare System, Rehabilitation, Research and Development, Boston, MA, USA. ✉e-mail: jaeikjq@gmail.com

conventional stimulation techniques. Encouragingly, ON and OFF cells have differential sensitivity to HFS[9,10]. OFF brisk-transient RGCs in rabbits were more sensitive to biphasic pulses delivered at 2 kHz and therefore could be preferentially activated at low amplitudes (matching natural responses to luminance decreases) while at higher amplitudes, activity in OFF cells was suppressed while ON responses become robust (matching natural responses to luminance increases)[9]. Variations in the frequency and amplitude of HFS may enable further selectivity between sustained and transient types of RGCs[10], thus offering the potential to reproduce other elements of natural signaling as well.

Further optimization of HFS is still required, however, as the level of selectivity for activation of individual cell types is still limited. The factors that mediate the HFS-induced suppression of spiking in RGCs or fibers are still not fully understood and previous modeling studies offer differing explanations. In the peripheral nervous system (PNS), some studies suggest that the inhibition of neuronal activity arises because voltage-gated potassium channels in the cell membrane become persistently activated by HFS, the resulting efflux of potassium overwhelms depolarizing sodium currents[28–31]. Other studies find that HFS causes tonic membrane depolarization that keeps voltage-gated sodium channels in the inactivated state (depolarization block)—the lack of available sodium channels prevents the generation of subsequent action potentials[15,32]. Similar to the PNS, one modeling study in the retina found that depolarization block mediated the suppression of spiking[33], but another retinal study found that depolarization block contributes only to the initial suppression of spikes in RGCs while subsequent hyperpolarization, caused by a reversal of the sodium current, is primarily responsible for the persistent inhibition during HFS[34]. The factors that mediate sensitivity differences between different types of neurons are also not well understood. Kameneva et al.[33] showed that higher concentrations of voltage-gated potassium channels lower the sensitivity to HFS in RGCs, raising the possibility that ON vs. OFF types of RGCs might express different levels of these channels. Meanwhile, Guo et al.[34] suggested that the variations in the expression level of voltage-gated sodium channels are likely to be a more important factor to the sensitivity differences across cell types.

The suppression induced by HFS has also been explored using bifurcation theory[35–37]. Bifurcation analysis allows the behavior of a dynamical system to be studied, particularly in understanding how a small parameter change can result in the qualitative shift, both in the case of single neuron models[38–43] and network models[44–46]. Bifurcation analysis on HFS has suggested that the suppression of action potentials in response to HFS is potentially related to the stabilization of the neuron's resting state of the averaged dynamics[35]. Another bifurcation study has proposed that HFS could lead to stabilization of unstable fixed points[36]. While these studies offer intriguing insights into the dynamic behavior of neurons to HFS, the physiological validation is still pending.

To investigate the factors shaping responses to HFS in RGCs, we used physiological measurements to evaluate the responses of individual cells in the explanted mouse retina to stimulus frequencies ranging from 100 to 2500 Hz using whole-cell patch clamp recordings. Measurements were restricted to ON and OFF α sustained RGCs; the use of a single type of RGC helped to avoid the confounding effects of response differences that can exist across types[9,29,47–50]. Use of whole-cell patch recordings is especially attractive for this type of work because it ensures that the electrical (stimulus) artifact, especially prevalent with HFS, does not obscure the neuronal responses and thus both the spiking response as well as changes to membrane voltage ($V_m$) could be captured and compared across stimulating conditions. We found that HFS causes shifts in baseline membrane potential and that both the strength and the polarity of the shift were dependent upon the frequency and amplitude of stimulation. Spiking patterns in each cell were directly correlated to the strength and polarity of the shifts in $V_m$. Consistent with previous findings in the rabbit retina, ON and OFF RGCs in mice had different sensitivities to HFS, and further, the changes in $V_m$ required to induce both activation and inactivation (depolarization block) were different in ON vs. OFF types. Development of a computational model allowed

the specific biophysical mechanisms mediating the observed responses to be probed.

## Results

### Response patterns of RGCs vary across stimulus frequencies

We recorded responses of ON α sustained RGCs to stimulus frequencies ranging from 100 to 2500 Hz and stimulus amplitudes ranging from 10–100 μA (Fig. 1); the duration of the stimulus train was fixed at 1 s. Measurements were restricted to a single type of RGC because much previous work has shown that responses to artificial stimulation can vary significantly across types[9,47–51]. Further, because recent studies suggest that even within a given cell type, both synaptic inputs as well as intrinsic properties can vary with retinal location[52–55], the experiments here were limited to ON and OFF α sustained cells from the temporal retina only.

Consistent with much previous work, the number of spikes elicited by relatively low frequencies of stimulation, e.g., 200 Hz, generally increased with increasing stimulus amplitude before plateauing[1] (referred to as monotonic responses); for the ON α sustained cell in Fig. 1a, the plateau occurred at a stimulus amplitude of 30 μA. At higher rates of stimulation, e.g., 1500 Hz, responses increased initially with increasing stimulus amplitude before reaching a maximum at an intermediate level (40 μA for the cell of Fig. 1c); further increases in stimulus amplitude led to a reduction in spike counts. These response patterns, referred to as non-monotonic, are consistent with previous studies of RGC responses to high-rate stimulation[9,10,50]. Interestingly, responses to intermediate frequencies of stimulation, e.g., 500 Hz, were non-monotonic for weaker stimulus amplitudes (0–50 μA) but then became monotonic as stimulus levels increased further (Fig. 1b).

Figure 1d maps the response across the full range of stimulus frequencies and amplitudes used in this study. The color of each pixel indicates the number of spikes elicited by a single combination of stimulus frequency and amplitude (color bar shown at right). Responses to a given stimulus frequency are contained within a single row with the amplitude of stimulation increasing from left to right. Consistent with the results of Fig. 1a–c, responses tended to increase monotonically with amplitude for low frequencies of stimulation while at higher frequencies, responses tended to be non-monotonic. At intermediate stimulus rates (400–700 Hz), responses were non-monotonic at low stimulus amplitudes but became monotonic as stimulus amplitude increased. For stimulus frequencies that elicited a monotonic response, the amplitude at which the response plateaued tended to increase with increasing stimulus frequency (plateau levels are indicated by a downward-pointing arrowhead, ▼). Also, the peak number of elicited spikes typically closely matched the applied stimulus frequency (i.e., ~300 spikes for 300 Hz), except for responses to 100 Hz which plateaued at levels of both ~100 and ~200 spikes (Supplementary Fig. 1). We did not observe monotonic responses for stimulus frequencies above 900 Hz. For rates that elicited non-monotonic responses, the amplitude that generated the strongest response is labeled with an asterisk (*) and increased slightly with increasing stimulus frequency although peak response levels remained largely constant.

Figure 1e, f shows the spiking response as a function of stimulus frequency for fixed current amplitudes of 40 and 70 μA, respectively (corresponding to the labels at the top of Fig. 1d). At low amplitudes, all stimulus frequencies elicited spikes (Fig. 1e) while at higher amplitudes, once the peak firing rate was achieved, there was a strong suppression of spiking that persisted for all frequencies tested, although there is some hint that spiking might have resumed if we had tested even higher stimulation frequencies (Fig. 1f).

### Sinusoidal stimulation leads to shifts in baseline membrane potential

The factors shaping non-monotonic responses to HFS were explored in a previous modeling study by Kameneva et al.[33]. They suggested that the level of membrane depolarization increased with the amplitude of stimulation (for high-rate stimulation of 2000 Hz), and excessive depolarization by strong stimulus amplitudes suppress spiking activities. Guo et al.[34] also used

**Fig. 1 | RGC response patterns show variation with stimulus frequency. a–c** The number of spikes elicited in an ON α sustained RGC from the explanted mouse retina is plotted as a function of current amplitude for stimulus frequencies of 200 (**a**), 500 (**b**), and 1500 (**c**). Arrowheads indicate the amplitude level at which monotonic responses saturated, and asterisks indicate the amplitude at which non-monotonic responses peaked. The duration of the stimulus was 1 s. **d** The number of spikes elicited in a representative cell for all combinations of frequency and amplitude is mapped. Consistent with (**a–c**), arrowheads and asterisks indicate plateau levels and peak responses, respectively. The stimulus amplitudes of 90–100 μA for 100 Hz (colored in gray) were not tested to prevent potential damage on stimulating electrodes and neurons by excessive electric charge. **e, f** The number of spikes is plotted as a function of stimulus frequency for current amplitudes of 40 (**e**) and 70 μA (**f**). In all plots, each combination of frequency and amplitude was applied at least five times. Error bars indicate the standard deviation.

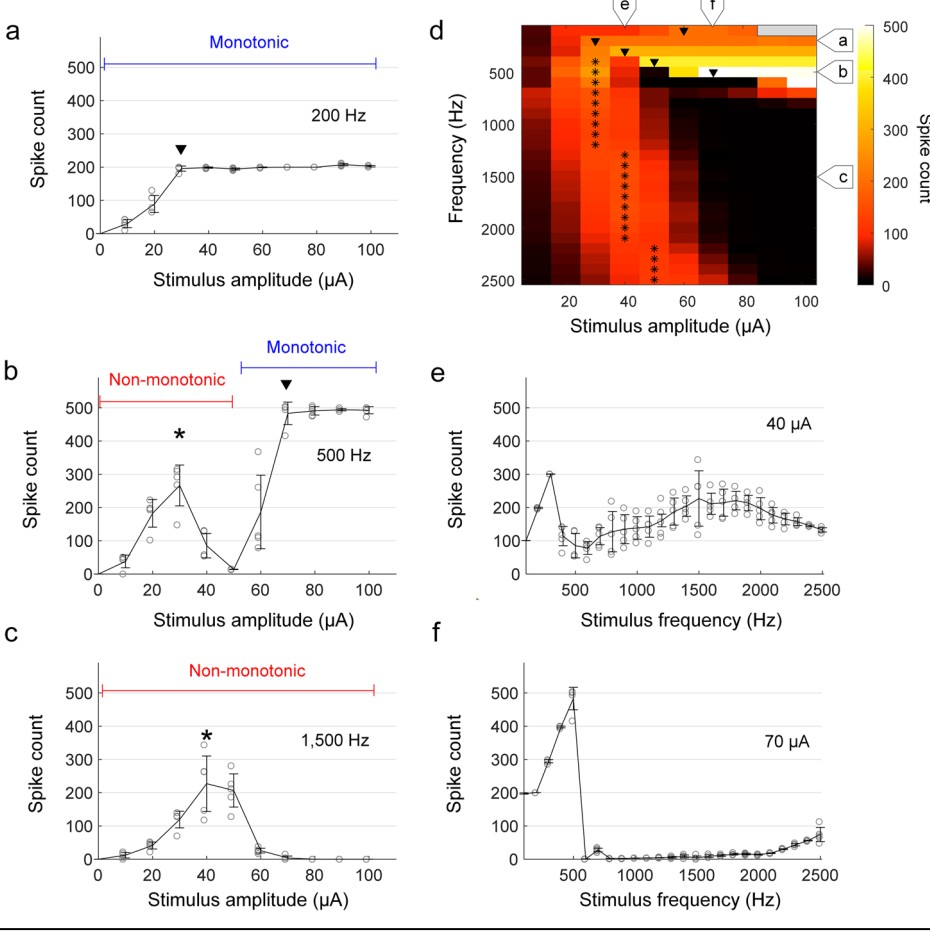

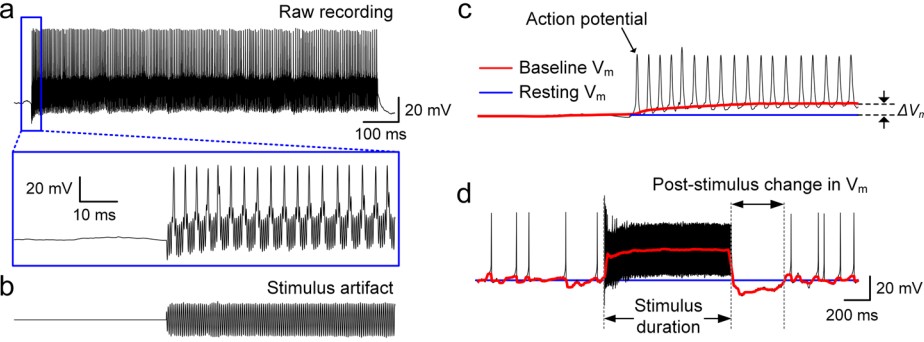

**Fig. 2 | Sinusoidal stimulation leads to changes in baseline membrane potential. a** An in vitro whole-cell recording of a representative RGC response to 2000 Hz sinusoidal stimulation (duration of 1 s and amplitude of 50 μA). An expanded view of the recording at stimulus onset (blue box) is shown below. **b** Stimulus artifact extracted from the whole-cell recording by fitting with a sinusoidal function. **c** After removing the stimulus artifact from the raw recording (i.e., **a**, **b**), the baseline membrane potential (red trace) is obtained by median filtering (see "Methods"). $\Delta V_m$ was quantified by the difference between resting membrane potential (i.e.,

membrane potential before stimulus onset; blue trace) and shifted baseline membrane potential during stimulation. The scale bar in the inset of (**a**) applies to both (**b**) and (**c**). **d** Strong HFS resulting in a substantial depolarization was followed by a momentary hyperpolarization below the resting level. This post-stimulus changes in the membrane potential typically lasted for less than 500 ms. Same as in (**c**), the red and blue traces indicate the baseline membrane potential and resting membrane potential level, respectively.

modeling to analyze responses to HFS and similarly reported that the level of membrane depolarization increased with stimulus amplitude. However, their model showed that the level of membrane depolarization exceeded the sodium reversal potential at strong stimulus amplitudes—the resulting inversion of the sodium current led to hyperpolarization of the membrane current, which suppressed spike generation[34]. Given that both modeling studies report that changes to the baseline membrane potential ($V_m$) play a key role in the suppression of spiking for HFS, we sought to measure the

changes in $V_m$ arising at different frequencies and amplitudes of HFS and to explore their influence on the response patterns.

Figure 2a shows the membrane voltage response to a 2000 Hz stimulus (see "Methods"). In the expanded time scale (bottom), the transition of the membrane potential from resting to regular spiking is clearly visible, which is also related to bifurcation (see Supplementary Fig. 2). In addition to the presence of action potentials, a change in $V_m$ occurred rapidly after stimulus onset. To quantify this shift, we digitally removed the electric artifact

**Fig. 3 | The change in baseline membrane potential varies with stimulus frequency. a–c** The average membrane potential changes ($\Delta V_m$) induced by stimulus frequencies of 200 (**a**), 500 (**b**), and 1500 Hz (**c**) are plotted as a function of the stimulus amplitude. **d** The heat map shows $\Delta V_m$ induced by each stimulus condition in a representative cell from the explanted mouse retina (the same cell in Fig. 1). The positive and negative signs in the legend indicate depolarization and hyperpolarization, respectively. **e, f** The level of $\Delta V_m$ is plotted as a function of stimulus frequency for current amplitudes of 40 (**e**) and 70 μA (**f**). In all plots, each combination of frequency and amplitude was applied at least five times. Error bars indicate the standard deviation.

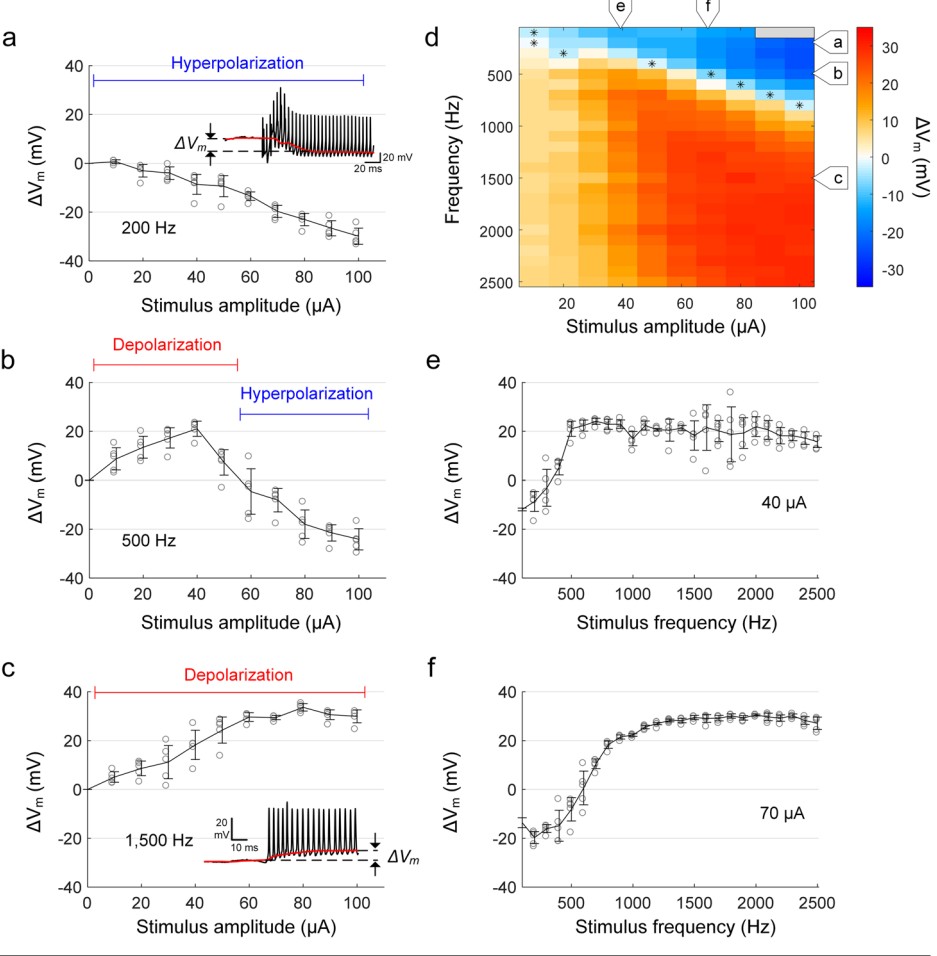

(Fig. 2b) from the raw recording (see "Methods") allowing the change in $V_m$ (referred to as $\Delta V_m$) as well as the individual action potentials to be more clearly visualized (Fig. 2c). Following the cessation of the stimulus, the membrane potential underwent a transient fluctuation and then returned to its original resting level, typically in less than 500 ms (Fig. 2d). We found that $\Delta V_m$ varied with both the frequency and amplitude of stimulation and could be positive (depolarization) or negative (hyperpolarization) (Fig. 3). For example, at 200 Hz, increases in the amplitude of stimulation led to increasing levels of hyperpolarization (increasingly negative values of $\Delta V_m$, Fig. 3a). In contrast, increases in the amplitude of stimulation at a frequency of 1500 Hz led to increasing levels of depolarization (positive $\Delta V_m$) that plateaued for amplitudes stronger than 60 μA (Fig. 3c). At intermediate frequencies of stimulation, e.g., 500 Hz, increasing the level of stimulation initially led to increasing levels of depolarization (Fig. 3b, up to 40 μA), but further increases resulted in hyperpolarizing changes of $V_m$. We measured $\Delta V_m$ for all combinations of stimulus frequency and amplitude and mapped the results (Fig. 3d) where $\Delta V_m$ levels are represented by different colors; warm (red) colors represent depolarization and cool (blue) colors represent hyperpolarization (color bar at right). For stimulation frequencies that induced hyperpolarizing changes in $V_m$, an asterisk is used to indicate the minimum stimulus amplitude for which hyperpolarization was observed. The resulting pattern suggests that hyperpolarization of $V_m$ persisted for increasing levels of stimulus amplitude as the frequency of stimulation increased although hyperpolarization could not be elicited for stimulation frequencies ≥1000 Hz, at least not for the range of stimulus amplitudes tested here. For stimulation frequencies ≥1000 Hz, depolarization was more prominent, especially for amplitudes ≥60 μA (red pixels in lower right quadrant of Fig. 3d).

Analogous to the plots in Fig. 1e, f, we plotted $V_m$ as a function of stimulus frequency for the same two amplitude levels (40 and 70 μA) (Fig. 3e, f). For each level of current amplitude, the shift in $V_m$ generally increased with stimulus frequency, plateauing at the highest frequencies. The sustained depolarization for higher stimulation frequencies at 70 μA (Fig. 3f) corresponds to the complete suppression of spiking (Fig. 1f) while for the lower stimulus amplitude of Fig. 3e, the level of $V_m$ depolarization was slightly less and was associated with sustained spiking (Fig. 1e).

The overall pattern of $\Delta V_m$ shown in Fig. 3d is largely similar in appearance to the pattern of spiking responses observed in the same cell as shown in Fig. 1d. For example, the region with high levels of depolarization (large red area at the bottom right of Fig. 3d) is similar in both size and shape to the region with no spiking responses (large black area at the bottom right of Fig. 1d). Similarly, the region with high levels of hyperpolarization (triangular blue region in the top right of Fig. 3d) is similar to the region with monotonic responses (triangular portion in the top right of Fig. 1d). Also, the same intermediate frequencies that exhibit both non-monotonic and monotonic spiking responses (Fig. 1d) exhibit both hyperpolarization and depolarization (Fig. 3d). Given these similarities, we compared results from the two heat maps in Supplementary Fig. 3. For each frequency, the blue line with arrowheads indicates the range of stimulus amplitudes that led to monotonic responses while the red line indicates the amplitudes that led to non-monotonic responses (Fig. 1d). Also, the blue shading behind each line indicates combinations of amplitude and frequency that hyperpolarized membrane potentials while red shading indicates depolarization of $V_m$ (Fig. 3d). The comparison shows that stimulus conditions that hyperpolarized $V_m$ also resulted in monotonic responses, while those that produced depolarization led to non-monotonic responses.

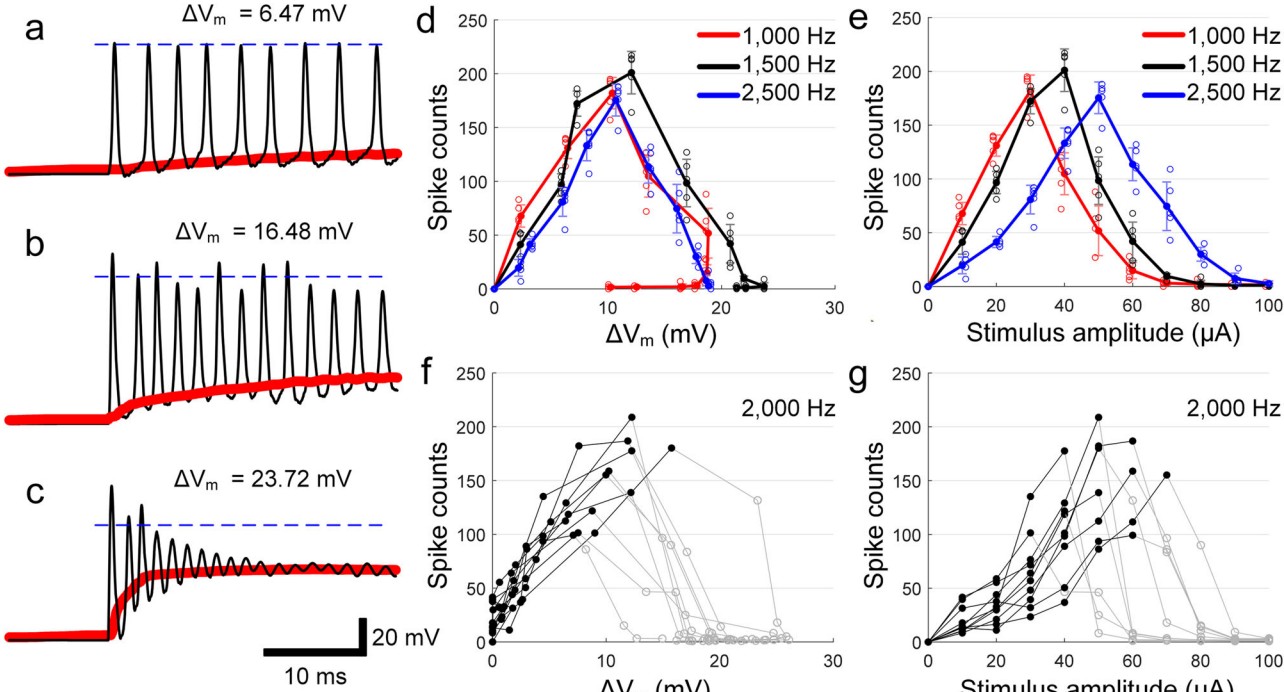

**Fig. 4 | Excessive depolarization leads to spike suppression. a–c** Typical in vitro whole-cell patch recordings (artifact-subtracted, see "Methods") at low, medium, and high levels of depolarization, respectively. Red traces show the shift in baseline membrane potential. Stimulus currents are 20 (**a**), 40 (**b**), and 60 μA (**c**). The dotted blue horizontal lines are arbitrarily positioned to help facilitate comparisons of spike amplitude. **d, e** The number of spikes is plotted as a function of $\Delta V_m$ (**d**) and current amplitude (**e**) for stimulation frequencies of 1000, 1500, and 2500 Hz. Each combination of frequency and amplitude was applied at least three times and error bars indicate the standard deviation. **f, g** The number of spikes is plotted as a function of $\Delta V_m$ (**f**) and current amplitude (**g**) in response to 2000 Hz stimulation. The black and gray line segments indicate portions of the curve below and above the peak firing rate, respectively. Error bars are omitted for clarity.

To further explore the relationship between $V_m$ and spiking, we re-examined the individual response traces for three different levels of membrane depolarization (Fig. 4a–c, black traces). For stimulus conditions that resulted in relatively low levels of depolarization (Fig. 4a), elicited spikes arose at uniform intervals and the amplitude of all spikes was approximately constant. For higher levels of depolarization, however, the amplitudes of spikes were less regular (Fig. 4b), and at even stronger levels of depolarization, peak amplitudes decreased rapidly over the course of stimulation (Fig. 4c), leading to the cessation of spiking. Interestingly, when spike count was plotted as a function of $\Delta V_m$ (Fig. 4d) the maximum spike count occurred at the same approximate level of $\Delta V_m$ regardless of the stimulus rate. This was not the case however when spike count was plotted vs. the amplitude of the stimulus current (Fig. 4e). Taken together, the results from Figs. 3 and 4 suggest that the level of $\Delta V_m$ induced by stimulation mediates the transition from monotonic to non-monotonic spiking, i.e., not the amplitude of the stimulus. Further support for the importance of $\Delta V_m$ comes from overlaying the plots of spike counts vs. induced $\Delta V_m$ across all cells for a given stimulus frequency (2000 Hz) (Fig. 4f): before spike counts begin to decrease, a strong linear relationship was observed between the level of $\Delta V_m$ and the resulting spike count (Pearson's $r$ value: 0.86). In contrast, there was a much broader distribution when the same dataset was plotted as a function of stimulus amplitude (Fig. 4g; Pearson's $r$ value: 0.66). The similarity of responses when plotted vs. $\Delta V_m$ strongly suggests responses to high-frequency stimulation are indeed dependent on the level of $\Delta V_m$ induced by a given set of stimulating conditions and not by the amplitude of stimulation or any other stimulation parameter. Thus, spiking responses remain robust to stimuli that lead to membrane hyperpolarization, regardless of the stimulus amplitude, while stimuli that lead to membrane depolarization typically result in a breakdown in firing (non-monotonic responses), especially as depolarization levels began to exceed ~10 mV.

## The variation in ON and OFF responses arises from their differences in depolarization sensitivity and block thresholds

Given that changes in membrane potential appear to underlie the pattern of non-monotonic responses to high-frequency stimulation, we next explored the factors that mediate the reported differences in sensitivity between ON and OFF types[9,10,50]. We applied sinusoidal stimulation at 2000 Hz to both ON ($n = 11$) and OFF ($n = 8$) α sustained cells and compared spiking responses (Fig. 5a–c) as well as induced membrane polarization (Fig. 5d–f) across the two populations. Although there was no statistical difference in the average peak spike counts between the two populations (Fig. 5b), on average the peak responses of ON cells occurred at lower stimulus amplitudes than those of OFF cells ($p < 0.001$, Fig. 5c). There were also significant differences in the $\Delta V_m$ levels at which elicited spikes began to decrease (compare red and blue circles in Fig. 5d) with ON cell responses remaining monotonic for higher levels of $\Delta V_m$ vs. OFF cells ($p = 0.013$, 95% CI [1.12, 8.18], Cohen's d value of 1.29, Fig. 5e). This is somewhat surprising given that the breakdown of responses occurred at lower stimulus amplitude in ON cells (Fig. 5a, c). It is also intriguing that the membrane potential of ON cells depolarized more than that of OFF cells for a given stimulus amplitude. We define sensitivity as the $\Delta V_m$ level at which breakdown occurs divided by the corresponding stimulus amplitude—the sensitivity was significantly higher in ON cells (0.27 and 0.14 mV/μA for ON and OFF cells, respectively; $p < 0.001$; Fig. 5f). Thus, even though the breakdown $\Delta V_m$ tended to be higher in ON cells (Fig. 5e), their higher sensitivity led to ON cells reaching their breakdown $\Delta V_m$ at lower stimulus amplitudes. These results therefore suggest that the difference in response curves between the two types of RGCs is shaped by both the transformation of stimulus strength into membrane depolarization by each cell type as well as by the level at which each enters depolarization block (Fig. 5a). These results were not dependent on use of a 1-s duration over which spikes were counted since cells typically entered depolarization block after only a few spikes following the onset of

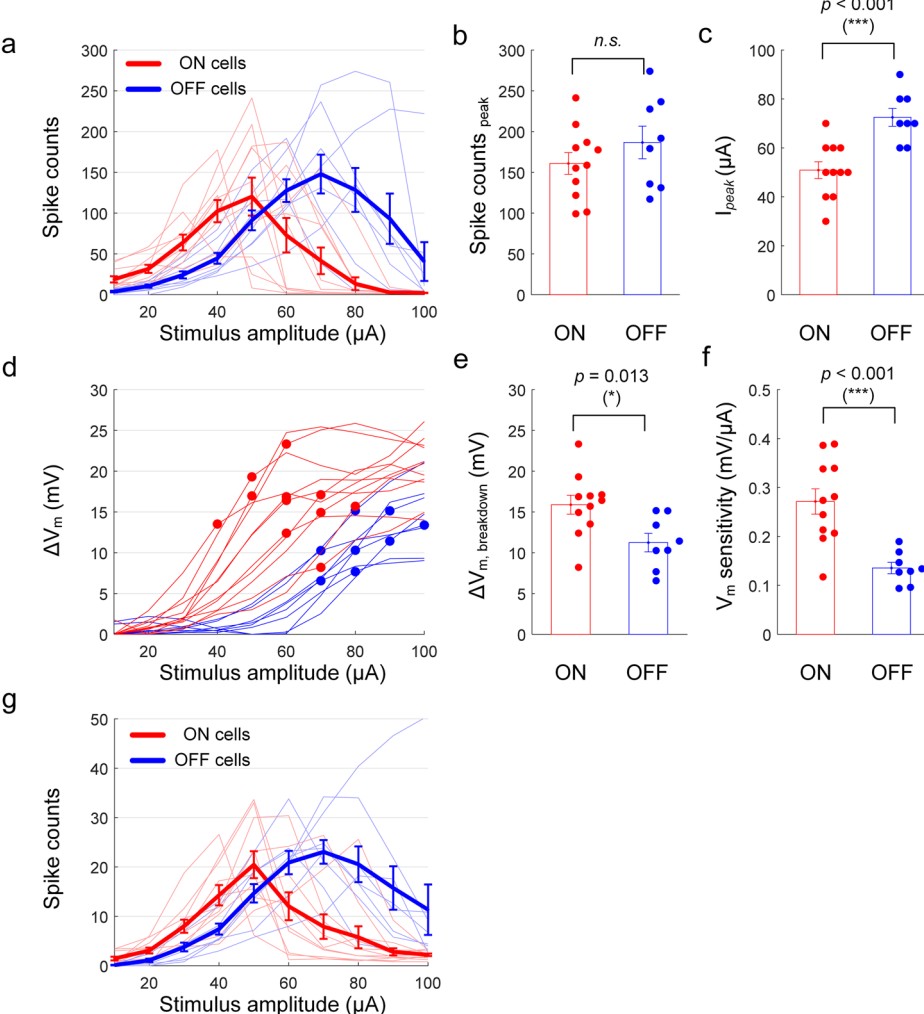

**Fig. 5 | The differences in the responses of ON vs. OFF RGCs to HFS arise from differences in their depolarization sensitivity as well as the voltage level at which depolarization block occurs in each type. a** The number of spikes elicited in ON ($n = 11$) and OFF ($n = 8$) cells from explanted mice retinas as a function of stimulus amplitude of 2000 Hz stimulation. Thin lines indicate the average per cell, and the thick solid lines indicate the average per population. Error bars indicate the standard error of the mean (SEM). **b, c** Comparisons of peak spike counts (**b**; $p = 0.279$, 95% CI [−22.87, 74.41], Cohen's d value of 0.52) and stimulus amplitudes at which peak responses were elicited, $I_{peak}$, (**c**; $p = 5.51e − 4$, 95% CI [10.85, 32.33], Cohen's d value of 1.97) between the ON and OFF cells. **d** The induced membrane potential change, $\Delta V_m$, as a function of stimulus amplitude. Each line indicates the average per cell, and the circles indicate the breakdown point where the elicited spikes begin to decrease. **e, f** Comparisons of breakdown $\Delta V_m$ (E; $p = 0.013$, 95% CI [1.12, 8.18], Cohen's d value of 1.29) and sensitivity (F; $p = 5.66e − 4$, 95% CI [0.07, 0.20], Cohen's d value of 1.96) between the ON and OFF cells. The sensitivity was defined by breakdown $\Delta V_m$ divided by the corresponding stimulus amplitude. **g** For the same dataset used in (**a**), the number of spikes elicited only during the first 100 ms post-stimulus onset were counted. Error bars indicate the standard error of the mean (SEM).

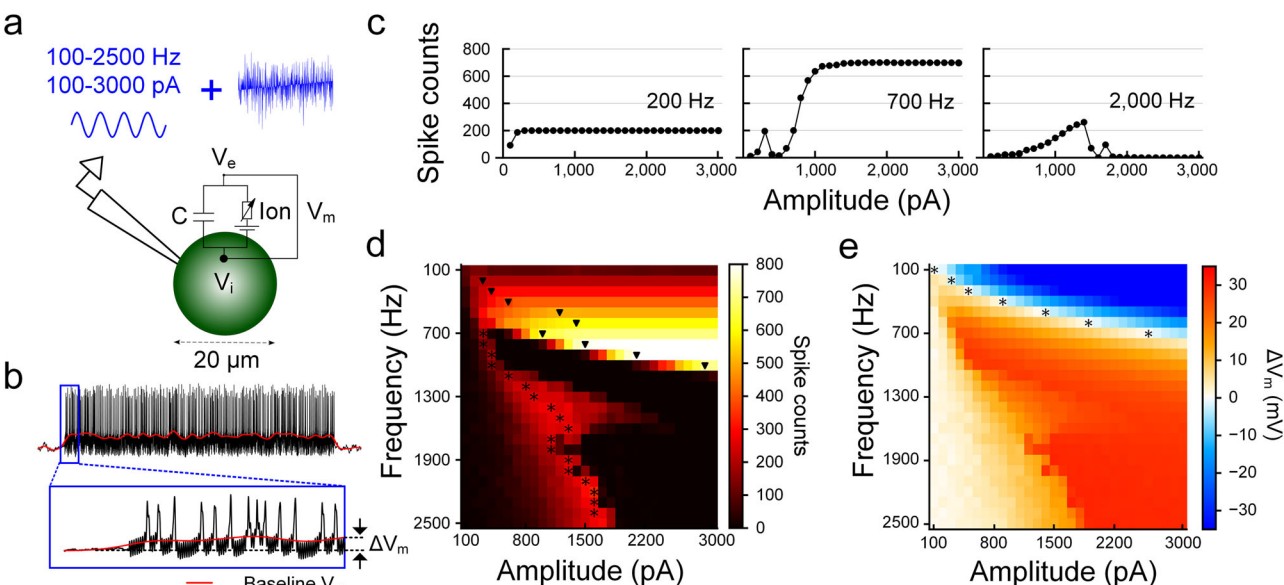

**Fig. 6 | Modeling results closely match the results from physiological experiments. a** Schematic of the stimulation and recording configuration as well as the equivalent electric circuit of the single-compartment model. Stimulation was applied intracellularly by injecting a sinusoidal waveform with the same frequencies used physiologically. **b** Representative trace of membrane potential over time in response to high-frequency stimulation. **c** Spiking response is plotted versus the amplitude of the injected current for stimulus rates of 200, 700, and 2000 Hz. **d, e** Heat maps showing spike counts (**d**) and $\Delta V_m$ (**e**) for all frequency and amplitude combinations tested. Color bars at right.

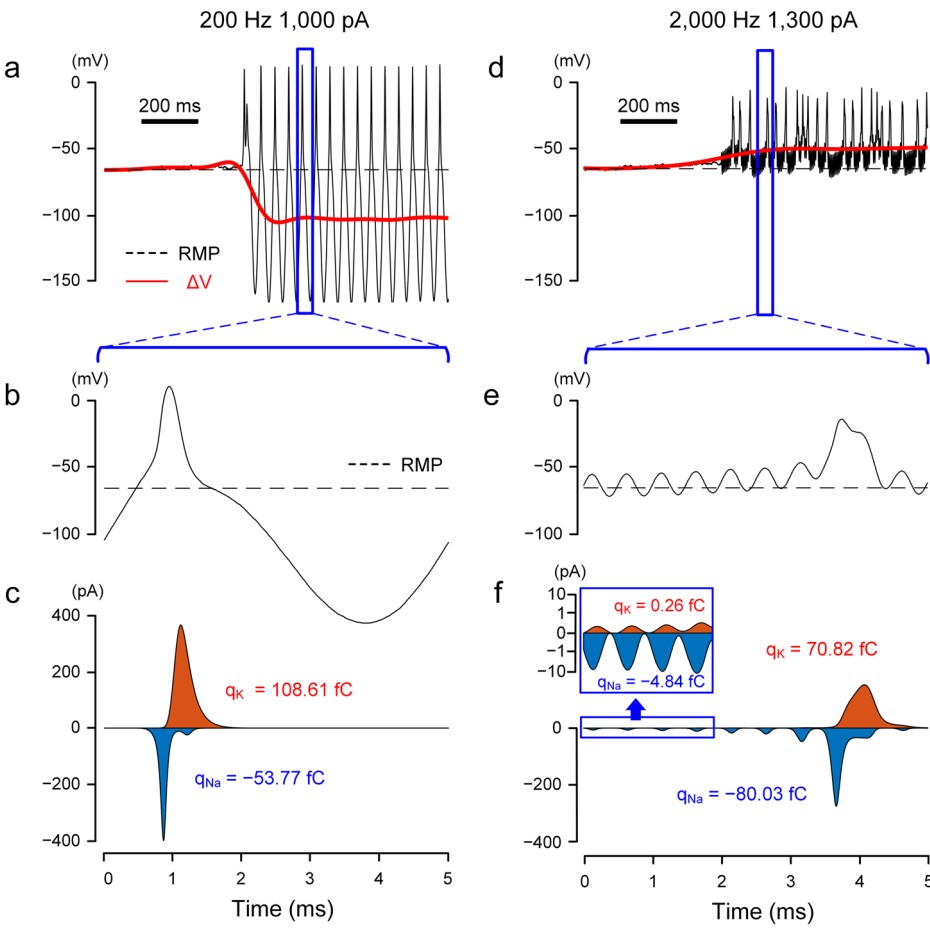

**Fig. 7 | Sodium and potassium channels have different sensitivities to low vs. high-frequency stimulation. a** Artifact-subtracted membrane potential (black) and $\Delta V_m$ (red) in response to 200 Hz stimulation. The dotted line indicates the resting membrane potential. **b** Expanded view of the membrane potential in (**a**), corresponding to one period of the sinusoidal waveform. **c** Sodium (blue) and potassium (orange) currents correspond to (**b**). The x-axis applies to both (**b**) and (**c**). The total charges transferred through the sodium ($q_{Na}$) and potassium ($q_K$) channels were calculated by the area under each curve. **d–f** Same as (**a–c**) but for 2000 Hz stimulation. (**e**) and (**f**) display the traces for the membrane potential and ion currents for 5 ms, corresponding to ten periods of the sinusoidal waveform. In the inset of (**f**), the logarithmic scale was used on the y-axis to visually accommodate both sodium and potassium currents of which peak values differ by more than one order.

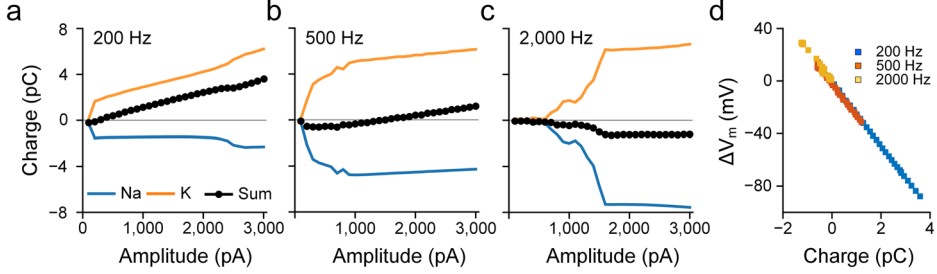

**Fig. 8 | Ionic current imbalance leads to differential membrane polarization during HFS. a** For 200 Hz stimulation, sodium (blue), potassium (orange), and the summed (black) charges were plotted as a function of stimulus amplitude. The charge was recorded for a period of 10 ms starting 500 ms after pulse onset. **b, c** Same as (**a**) but for 500 (**b**) and 2000 Hz (**c**). **d** $\Delta V_m$ induced by each stimulus amplitude of 200, 500, and 2000 Hz stimulation is plotted as a function of the summed charge (Pearson's $r$ value: 0.99).

stimulation (Fig. 4c) and therefore ON vs. OFF response curves for the first 100 ms post-stimulus onset (Fig. 5g) were similar to those for 1 s (Fig. 5a).

## Ion channel dynamics underlie membrane polarization

To better understand how $\Delta V_m$ influences spiking, we developed a computational model that allowed specific biophysical properties of the cell to be studied in isolation (see "Methods"; Fig. 6a, b). The simplified model consisted of a single spherical compartment (20 μm in diameter) with membrane dynamics that closely followed previously published model RGCs[56]. Stimulation was delivered by injecting current into the cell (100–3000 pA) at the same stimulus frequencies used in physiological experiments (100–2500 Hz). Somewhat surprisingly, responses in the single-compartment model cell demonstrated a qualitative agreement with physiological testing, e.g., monotonic responses for low frequencies of

stimulation, non-monotonic responses for high frequencies and combination of both for intermediate frequencies (Fig. 6c). Analogous to the approach with physiological measurements, two heat maps were generated: the first mapped the number of spikes elicited across stimulus frequencies and amplitudes (Fig. 6d) while the second mapped $\Delta V_m$ across the same two parameters (Fig. 6e). Although there are slight differences between the physiological maps and the maps from the model, the two model-elicited maps correspond well to one another, i.e., the differences between the $\Delta V_m$ heat maps for physiology vs. the model are similar to the differences between the spiking heat maps for physiology vs. model. The only significant difference between the model cell and physiological experiments was that $V_m$ was more strongly hyperpolarized during low-frequency stimulation (top right corner in Fig. 6e). We did not investigate this discrepancy but speculate that it is due to not including one or more voltage-gated ion channels in the

**Fig. 9 | The inactivation of the sodium channel underlies the depolarization block. a** Transition into depolarization block during 2000 Hz stimulation. The black trace represents the membrane potential after artifact-subtracted, while the red line indicates the baseline membrane potential ($\Delta V_m$). **b** Corresponding sodium $m$ and $h$ gating variables (blue and orange, respectively) are plotted over time.

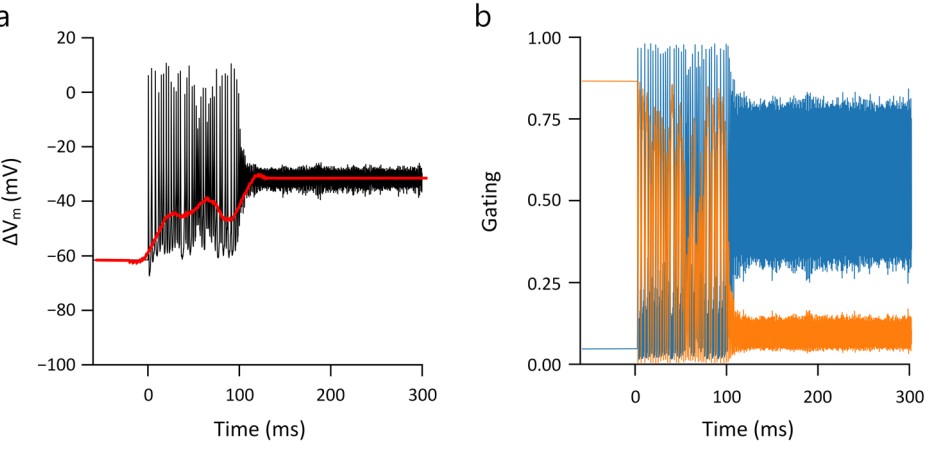

**Fig. 10 | Variation in non-monotonic response curves depending on modeling parameters. a–c** The spiking responses to 2000 Hz stimulation were simulated using a single-compartment model. While maintaining other parameters consistent with those in Fig. 6, we investigated the impact of changes in cell size (**a**), potassium channel density (**b**), and noise level (**c**) on the response curves. Blue traces indicate standard parameters.

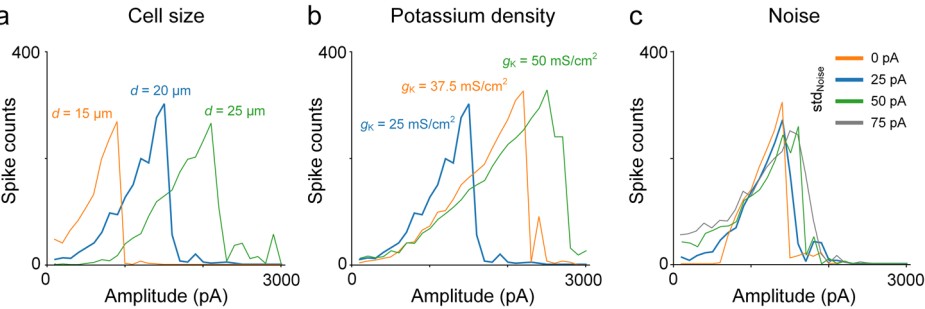

model—e.g., hyperpolarization-activated cyclic nucleotide-gated (HCN) channels which repolarize $V_m$ during hyperpolarization[57,58]. Regardless, the fact that results from the single-compartment model demonstrated a qualitative agreement with physiological results suggests that the depolarizing and hyperpolarizing effects arise intrinsically, e.g., from the kinetics of the ion channels in the cell membrane, and not from features of the somato-dendritic and/or axonal morphology.

We investigated the specific biophysical factors mediating the shifts in $V_m$ and started by comparing the dynamics of voltage-gated ion channels for stimulation conditions that resulted in hyperpolarization (e.g., 200 Hz; Fig. 7a, b). Since sodium (Na) and potassium (K) currents are the principal driving forces that depolarize and repolarize the membrane potential, we extracted the Na and K currents and plotted them over the time course of the same stimulus (Fig. 7c). Since the kinetics of voltage-gated sodium ($Na_v$) channels are significantly faster than voltage-gated potassium channels ($K_v$), inward Na currents dominated the initial charge transfer (Fig. 7c), leading to a strong initial depolarization of $V_m$ (Fig. 7b). The rapid inactivation of sodium channels, coupled with the delayed onset of potassium outflux, results in the ensuing hyperpolarization of $V_m$. $K_v$ channels do not similarly inactivate rapidly and therefore remain open until $V_m$ is returned to resting by the stimulus waveform and ion transfer (Fig. 7b, c). Because of this prolonged activation of potassium currents, the total charge arising from $K^+$ currents is larger than that for $Na^+$ currents (Fig. 7c), resulting in a net hyperpolarization of $V_m$. At higher frequencies, the depolarizing and hyperpolarizing phases are shorter (Fig. 7d, e), limiting the duration for which $K_v$ channels remain open. The combination of the slow kinetics of $K_v$ channels along with the short activation window (i.e., short depolarizing phase) leads to much smaller outward K currents relative to the inward Na currents (Fig. 7f), resulting in a net depolarization of $V_m$ (Fig. 7d, e).

To further explore the relationship between $V_m$ and the transfer of $Na^+$ and $K^+$ ions, we computed the total charge from Na (blue) and K currents (orange) as well as the sum (black) for different combinations of frequency and amplitude (Fig. 8). For 200 Hz, increasing stimulus amplitude led to

increases in the level of net outward charges (Fig. 8a). i.e., a gradual increase in the level of membrane hyperpolarization as a function of stimulus amplitude, which is in-line with $\Delta V_m$ observed physiologically during stimulation at frequencies below 500 Hz (Fig. 3a, d). A stimulation frequency of 500 Hz led to a net inward charge (i.e., depolarization) at low amplitudes but net outward charge (hyperpolarization) at higher amplitudes (Fig. 8b), reflecting the change from hyper- to depolarization of $V_m$ seen in the physiological experiments (Fig. 3b). Stimulation at 2000 Hz was associated with negative membrane charge for all amplitudes tested indicating membrane depolarization (Fig. 8c). We plotted the relationship between $\Delta V_m$ and the net membrane charge for the three frequencies tested and found a strong linear correlation between the two (Fig. 8d; $r^2 = 0.99$). Taken together, our results demonstrate that the difference in time constants between $Na_v$ and $K_v$ channels plays a key role in the bimodal shift of $\Delta V_m$. In this study, we employed the Fohlmeister model[56], which is widely used for retinal ganglion cells (see "Methods"). Although the specific properties of each ion channel of the actual neurons and model cells may not be identical, the fundamental difference in time constants (i.e., the faster activation and inactivation of $Na_v$ channels vs. $K_v$ channels) remains consistent for the neurons that generate action potentials. Therefore, it can be inferred that a similar bimodal shift in $\Delta V_m$ would be observed in other spiking cells throughout the central and peripheral nervous systems.

During the transition from regular spiking to depolarization block in the modeled responses (Fig. 9) membrane currents changed abruptly from large deflections, reflecting the repeating cascade of sodium influx followed by potassium outflux that occurs during action potentials, to small oscillations that result in an almost flat time course of the membrane potential. Examination of the gating variables for voltage-gated sodium channels revealed the underlying reason for the depolarization block to be the inability of the inactivation gate of the Na channel ($h$) to de-inactivate during high-frequency/high-amplitude stimulation (Fig. 9b), therefore, the driving force of sodium ($m^3 \times h$) was diminished, suppressing spike generation. This finding that depolarization block underlies the non-monotonic responses of

RGCs is consistent with the previous modeling prediction from Kameneva et al.[33].

## Intrinsic property differences can mediate the ON/OFF sensitivity differences

To evaluate whether sensitivity differences between ON and OFF cells would emerge from changes to the intrinsic properties within the single-compartment model, we ran an additional series of simulations in which the size of the soma was varied while all other parameters were held constant (Fig. 10a); in an analogous set of experiments, we then varied the conductance of potassium channels with all other parameters constant (Fig. 10b). In both cases, the amplitude level at which depolarization block occurred was sensitive to changes in the individual parameters. These results therefore support the notion that intrinsic differences between types can mediate sensitivity differences to HFS for different cell types although we cannot rule out the possibility that other factors contribute as well, e.g., multiple sub-types of voltage-gated potassium (and other) voltage sensitive channels can be expressed at multiple locations within individual neurons, and these may interact in complex ways that are difficult to replicate in a single-compartment model. Thus, while our modeling results suggest that intrinsic morphological and biophysical features do indeed shape the responses to HFS, further studies with more realistic neuronal models are needed before making more definitive conclusions about the mechanism(s).

## Does baseline firing influence sensitivity to HFS?

In the retina, conditions like retinitis pigmentosa and age-related macular degeneration alter the synaptic inputs to RGCs, leading to increases in the rate of spontaneous firing[59,60]. Because these diseases lead to blindness, and thus may require a retinal prosthesis, we questioned whether response sensitivity to HFS is altered by changes in baseline firing levels by performing one additional set of simulations. We modified the noise level parameters in our model to increase the spontaneous firing rates but found only minor changes in the responses to HFS (Fig. 10c). These results are consistent with a previous study in RGCs that showed that the spike generation mechanism remains largely intact in the presence of the increased background spiking that occurs during retinal degeneration[61–63]. While these results therefore suggest that the sensitivity to HFS may remain consistent in the diseased retina, it will be important to directly assess efficacy of HFS in the degenerate retina given that the single-compartment model used here does not encompass the full spectrum of synaptic and intrinsic properties that change during retinal degeneration.

## Discussion

The ability of high-frequency stimulation (HFS) to inhibit neuronal activity, as demonstrated through conduction block in the PNS or through the suppression of spiking in RGCs[4,6,7,9,10,14–16], offers the potential for more effective control of neural activity and thus could lead to new medical treatments or enhanced performance of existing ones. To further develop this potential, however, it is important to better understand the biophysical mechanism(s) by which stimulation suppresses neural activity. Previous modeling studies have yielded conflicting results, with some suggesting conduction block is caused by sustained depolarization of membrane voltage[15,32,33] (referred to as depolarization block) while others suggest strong hyperpolarizing currents prevent firing[28–31,34]. Here, we directly measured the change in membrane voltage ($V_m$) in individual RGCs arising in response to sinusoidal stimulation with frequencies ranging from 100–2500 Hz and found that both the polarity as well as the magnitude of the shift in $V_m$ were dependent on the frequency and amplitude of stimulation. More specifically, any combination of stimulus parameters that caused the depolarizing change in $V_m$ to exceed a certain threshold level initiated depolarization block. In general, higher frequencies produced stronger depolarization, especially for increasing amplitudes of stimulation (Fig. 3). Thus, our results provide strong support for the notion that HFS-induced inhibition of spiking arises from depolarization block.

Results from our computational model demonstrated a qualitative agreement with physiological experiments and revealed that prolonged inactivation of $Na_v$ channels during continued depolarization (depolarization block) contributes to the cessation of spiking (Fig. 9). Interestingly, the model consisted of only a single compartment and therefore, the similarity to physiological results suggests the intrinsic dynamics of voltage-gated ion channels are largely responsible for the biphasic shifts in $V_m$.

Many aspects of the HFS responses observed here are similar to those in earlier research on both the retina[9,10] and PNS[2,64,65]. For example, consistent with previous studies[2,9,10,64,65], we observed that HFS typically suppressed spiking at frequencies above ~1 kHz (Fig. 1d). In addition, the observed rapid onset of suppression and swift return to the pre-stimulation state (upon initiation and cessation of HFS, respectively; Fig. 2d) are in alignment with previous reports[2,64,65]. It suggests that the primary factor for HFS-induced inhibition in our study (depolarization block), is likely to underlie similar suppression observed in other studies as well.

Interestingly, the level of $V_m$ at which depolarization block occurred was different for ON vs. OFF types of alpha sustained RGCs: depolarization block occurred for a $\Delta V_m$ of $15.89 \pm 3.86$ mV in ON cells vs. $11.24 \pm 3.20$ mV for OFF cells (Fig. 5d, e). The more depolarized level in ON cells was somewhat surprising since their peak firing rates tended to occur at lower stimulus amplitudes than in OFF cells (Fig. 5a, c). To better understand this apparent discrepancy, we compared the level of depolarization that arose in ON vs. OFF cells as a function of the amplitude of HFS and found that a given increase in amplitude produced a larger depolarization in ON (vs. OFF) cells (Fig. 5f). Thus, even though depolarization block does not occur in ON cells until the cell is more depolarized, the sensitivity of $\Delta V_m$ to stimulus strength is higher in ON cells and thus the stimulus amplitude at which the onset of depolarization block occurs is lower in ON vs. OFF cells. It is important to emphasize that our results do not demonstrate unequivocal preferential activation, nor do they suggest that HFS enables selectivity of all RGC types. However, the insights into the biophysics underlying complex and unique responses to HFS presented here provide an improved foundation for refining stimulation strategies with retinal prostheses, as well as for other neural engineering applications.

Our results do not reveal the causes for the observed differences in depolarization characteristics (i.e., breakdown $\Delta V_m$ and depolarization sensitivity; Fig. 5e, f) between ON and OFF alpha RGCs. However, given that responses of RGCs to HFS are not sensitive to blockers of synaptic input[9,10,47,50], these physiological response differences are likely to arise from differences in the intrinsic properties between the two cell types. Consistent with this, we found that changes to the size of the cell could shift the level at which the cell enters depolarization block (Fig. 10a), a finding that is consistent with earlier work in hippocampus and neocortical neurons[66]. We also found that changes to the expression levels of the voltage-sensitive K+ channels could also influence sensitivity to depolarization block, a finding similar to that from Kameneva et al.[33]. Furthermore, a previous study indicated different expression levels of voltage-gated calcium in ON vs. OFF RGCs[67], raising the possibility that multiple factors could combine to shape sensitivity differences between types. It will be interesting in future studies to learn the full extent of differences across these and other types of RGCs, and then to explore how best to harness such differences in the quest for improved selectivity with HFS.

Physiological and computational results from the present study as well as previous work[9,10,47,50] suggest that intrinsic properties of the neurons are largely responsible for the HFS-induced suppression in RGCs. This is also likely to be the case in the PNS where somas are too distant to be activated by the stimulus and there are no synaptic connections elsewhere along the axon. While there is some inclination to generalize our findings and conclude that HFS-induced suppression is mediated intrinsically within targeted neurons (or axons), it is important to note that the contribution of synaptic input to HFS-induced suppression (or other mechanisms besides depolarization block) may be more considerable in other neuronal systems. For example, in the neocortex and hippocampus, synaptic transmission plays a critical role in paroxysmal depolarization and depolarization block of

the cells, and depending on cell type (e.g., excitatory vs. inhibitory neurons) or size, the neurons have different depolarization block properties mediated by synaptic input[66]. Given their high depolarization block sensitivity to synaptic input, the HFS-induced suppression in these neurons might result from synaptic transmission triggered by HFS, with or without a contribution from the intrinsic ion channel dynamics of the cell membrane.

It is also important to note that our physiological results were exclusively obtained from explanted mouse retinas, and thus other mechanisms besides depolarization block could also be relevant in in-vivo conditions or other neuronal systems.

In addition to our primary focus on HFS responses, we also made several intriguing observations from the responses to low-frequency stimulation (LFS; <1000 Hz). LFS that produced stronger hyperpolarization of $V_m$ also produced a larger number of spikes (Figs. 1d and 3d), i.e., stronger spiking responses arose from hyperpolarized $V_m$ (not depolarized). We did not directly investigate the reason(s) for this, but speculate that hyperpolarization of $V_m$ produces a larger number of available (de-inactivated) voltage-gated sodium channels, which in turn better ensures the generation of spiking during each cathodal phase of sinusoidal stimulation. Another interesting observation was that ON and OFF α sustained RGCs show different sensitivities not only to HFS, but also to lower frequencies. For example, in response to 500 Hz, the peak non-monotonic responses occurred at weaker stimulus amplitudes in ON cells vs. OFF cells (compare the blue and red asterisks in Supplementary Fig. 4a). Furthermore, at the transition point where non-monotonic responses shifted to monotonic responses (indicated by the filled circles in Supplementary Fig. 4a), spike counts were considerably smaller in ON cells compared to OFF cells (10.7 ± 2.6 spike/s vs. 62.9 ± 20.43 spike/s). The two cell types also showed different depolarization sensitivity; weak stimulus amplitudes induced higher depolarization in ON cells compared to OFF cells (Supplementary Fig. 4b). While the small sample sizes for both cell types in Fig. 10b, c impose limitations on conducting statistical analysis, these results nevertheless suggest that LFS may be potentially useful for selective activation.

## Methods
### Preparation of retina
The care and use of animals followed all federal and institutional guidelines and all protocols were approved by the Institutional Animal Care and Use Committee (IACUC) of the Massachusetts General Hospital. Wild-type mice of the *C57BL/6J* strain (either sex) were purchased from the Jackson Laboratory (Bar Harbor, ME, USA) and sacrificed at the postnatal days ranging from 28 to 56. For euthanasia, animals were anesthetized via inhalation of vaporizing isoflurane, and then cervical dislocation was performed. To identify the orientation (i.e., dorsal, ventral, temporal, and nasal poles), a mark was made at the dorsal cornea midway between two canthi. After the eyeball was enucleated from the mice, a cut was made from the marker at the cornea toward the optic disk—this cut served to mark on the retina the orientation of the dorsal pole. After the retina was separated from the retinal pigment epithelium, it was mounted photoreceptor side down on a 10 × 10 mm² square piece of Millipore filter paper (0.45 µm HA membrane filter; EMD Millipore, Billerica, MA, USA) immobilized on a slide glass. A small hole (~2 mm in diameter) of the filter paper allowed light stimuli to be presented from below.

### Electrophysiology and light response
We used a whole-cell patch clamping technique to record spiking activities of RGCs. A patch pipette was filled with intracellular solution which consisted of (in mM, all Sigma-Aldrich): 125 K-gluconate, 10 KCl, 10 Hepes, 10 EGTA, 4 Mg-ATP, 1 Na-GTP. After break-in, the pipette series resistance was compensated with the bridge balance circuit of the amplifier (<15 MΩ). Recorded membrane potential was corrected for the change in liquid junction potential (−8 mV). Two silver chloride-coated silver wires with a diameter of 0.25 mm served as the ground. Each wire was separated by ~10 mm and was positioned ~15 mm away from a targeted cell. Retina

tissue was continuously superfused with Ames medium at ~36 °C (flow rate: ~4 mL/min), equilibrated with 95% oxygen and 5% carbon dioxide.

The experiments here were limited to ON and OFF α sustained cells from the temporal retina. After making a hole (typically 100–200 µm in diameter) in the inner-limiting membrane of the temporal retina, αRGCs were targeted based on their large somata (> 20 µm)[68,69], and then further classified by their responses to light stimuli[68,69]. Light stimuli were projected onto the outer segments of the photoreceptors from below using an LCD projector (PowerLite 760HD; Seiko Epson, Suwa, Nagano, Japan). Stationary flashes with diameters ranging from 100 to 1000 µm were presented for 1 s. Only cells showing consistent ON or OFF sustained responses across the full range of spot sizes were targeted for further analysis. Also, moving bars were presented to all cells to exclude directionally selective cells[70]. We collected and analyzed responses from 11 ON α sustained cells and 8 OFF α sustained cells.

Throughout the recording process, the resting membrane potential, the amplitude of the depolarizing peaks in action potentials, and spontaneous firing rate were regularly monitored. We also validated performance stability in individual neurons by regularly measuring the response to a reference or control stimulus condition in that cell. If a significant change was observed in any of these criteria, additional recordings were paused until values returned to their initial baseline levels. If the return to baseline levels did not occur, recordings were terminated in that cell.

### Data acquisition
Data were sampled at 10 kHz by NI-DAQ (NI PCI-MIO-16E-4; National Instruments, Austin, TX, USA) after low-pass filtering at 2 kHz using a MultiClamp 700B amplifier (Molecular Devices, Sunnyvale, CA, USA). The data acquisition was controlled by custom software written in LabVIEW (National Instruments, Austin, TX, USA) and MATLAB (MathWorks, Natick, MA, USA).

### Electric stimulation
Electric current was epiretinally delivered using a 10 kΩ platinum-iridium electrode (MicroProbes, Gaithersburg, MD, USA); the exposed area at the electrode tip was conical with an approximate height of 125 µm and base diameter of 30 µm, giving a surface area of 5900 µm². Its tip was located 25 µm above the targeted soma.

In clinical applications and previous studies, high-frequency stimulation has typically been delivered by a series of biphasic pulses or sinusoidal waveforms. In this study, which aimed to study frequency-dependent effects, we chose the sinusoidal waveform due to its discrete and focused frequency spectrum. In addition, the use of sinusoidal waveforms facilitated the removal of stimulus artifacts from the raw data (see the "Data analysis" section), thereby enabling a more precise quantification of spiking activity and the membrane potential. The sinusoidal stimulation was applied to target cells for a duration of 1 s. The frequency of sinusoidal waveform ranged from 100 to 2500 Hz, while the peak amplitude typically ranged from 10 to 100 µA (the maximum amplitude for 100 Hz stimulation was limited to 80 µA to prevent damage on stimulating electrodes and neurons by excessive charge). Stimuli were programmed and delivered by STG2004 system (Multi-Channel Systems MCS GmbH, Reutlingen, Germany). Two silver chloride-coated silver wires served as the return for the stimulating electrode.

Synaptic inputs were not blocked pharmacologically during our investigation of RGC responses to HFS. While blocking such input is well established for RGCs, such experiments would have necessitated holding the whole cell patch on each cell for an additional 20–30 min. Such an extension would have significantly limited the time available for exploring both the spiking and membrane voltage responses over the large parameter space used in this study. Furthermore, given much previous work showing that RGC responses to HFS are not sensitive to synaptic blockers[9,10,47,50], we chose to focus on pursuing new results rather than re-testing the effects of synaptic isolation.

## Data analysis

Stimulus artifact was removed from raw recordings by fitting with a sinusoidal function. For example, to remove artifacts from responses to 2000 Hz sinusoidal stimulation (Fig. 2a), a sinusoidal function with matching frequency, phase, and peak-to-peak amplitude was computed (Fig. 2b) and subtracted from the raw data (Fig. 2c). Individual spikes were identified as depolarization peaks and counted during the 1-s time window over which stimulation was applied. Spike amplitudes varied in some recordings, especially ones corresponding to the falling phase of non-monotonic responses (Fig. 4c). In those recordings, depolarization peaks with amplitude smaller than 20% of the largest peak (typically the first spike) were excluded from spike counting. The trace of the base membrane potential ($V_m$) was computed by median filtering the recordings (after artifact removal; a bin size for the median filtering was 40 ms; red solid line in Fig. 2c). The change in base membrane potential induced by stimulation ($\Delta V_m$) was defined by the difference between resting membrane potential (i.e., membrane potential before stimulus onset) and shifted baseline membrane potential at the saturation level during stimulation.

It should be noted that any harmonic transmembrane currents synchronized with the stimulus frequency would have been eliminated along with the artifact. However, we do not believe that this would have significantly affected our quantification of spiking activity and Vm. Action potentials have a characteristic shape that is distinguishable from the sinusoidal function, so removing the sinusoidal artifact would not have obscured our counts of elicited spikes. In addition, any harmonic transmembrane currents matching the stimulus frequency range (100–2500 Hz) would have been filtered out by the median filter during the calculation of Vm. Thus, even if these currents were inadvertently removed along with the artifact sinusoidal artifact, they would not have affected our quantification of $V_m$.

## Computational modeling

In order to analyze the basic response properties of RGCs during HFS, we developed a simple computational model in NEURON[71]. The spherical single-compartment model had a diameter of 20 μm and was equipped with membrane dynamics from Fohlmeister et al.[56]; ion channel densities and gating rate constants were set as follows:

**Ion channel densities (mS/cm²)**

$$g_{Na} = 50,\ g_K = 25,\ g_{Ca} = 1,\ g_{K,Ca} = 0.025,\ g_L = 0.2.$$

**Na activation**

$$\alpha_m(V) = -2.13\,(V + 35)/(e^{-0.1(V+35)} - 1)$$

$$\beta_m(V) = 70.97 e^{-(V+60)/20}$$

**Na inactivation**

$$\alpha_h(V) = 1.42 e^{-(V+52)/20}$$

$$\beta_h(V) = 21.29/(e^{-0.1(V+22)} + 1)$$

**K activation**

$$\alpha_n(V) = -0.076\,(V + 37)/(e^{-0.1(V+37)} - 1)$$

$$\beta_n(V) = 1.52 e^{-(V+47)/80}$$

**Ca activation**

$$\alpha_c(V) = -1.06\,(V + 13)/(e^{-0.1(V+13)} - 1)$$

$$\beta_c(V) = 35.48 e^{-(V+38)/18}$$

These values were originally taken from experimental data recorded from retinal ganglion cells[56]. Specific membrane capacitance was 1 μF/cm² and model temperature was set to 30 °C, similar to experimental conditions. Time step was set to 1 μs. Stimulation was applied intracellularly by injecting a sinusoidal current for 1 s with frequencies ranging from 100–2500 Hz and amplitudes ranging from 100–3000 pA. In addition, a simple white noise current was injected into the cell (mean = 0 pA, standard deviation = 25 pA, changed every 0.5 ms).

## Statistics and reproducibility

Linear correlation between two variables (e.g., spike counts vs. $\Delta V_m$, and spike counts vs. stimulus amplitude; Fig. 4f, g) was computed using Pearson's correlation coefficient. Unless otherwise indicated, all data were presented as the mean ± standard deviation. Statistical significance was verified using a two-sample t-test. Significance levels were set as follows: n.s. (not significant) $p \geq 0.05$, $*p < 0.05$, $**p < 0.01$, $***p < 0.001$. The effect size was determined using Cohen's d[72]. Reproducibility was ensured by conducting a minimum of three replicates. Information on sample sizes is provided in the corresponding figure legends.

## Reporting summary

Further information on research design is available in the Nature Portfolio Reporting Summary linked to this article.

# Data availability

The source data from physiological experiments and computational modeling are available on the Open Science Framework (https://doi.org/10.17605/osf.io/hnpq4).

# Code availability

Custom program codes and software used for computational analysis (Figs. 6–10) are also available on the Open Science Framework (https://doi.org/10.17605/osf.io/hnpq4)[73].

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

## Acknowledgements
This work was supported by National Institute of Neurological Disorders and Stroke (NINDS) of the National Institutes of Health (NIH; R01- NS110575), Congressionally Directed Medical Research Programs (CDMRP) funded by the Department of Defense (DOD; VR170089), National R&D Program through the National Research Foundation of Korea (NRF) funded by the Ministry of Science and ICT (2022M3E5E8017395), and Austrian Science Fund (FWF P35488).

## Author contributions
Jae-Ik Lee: Conceptualization, physiological experiment, data analysis, methodology, visualization, writing—original draft. Paul Werginz: Computational modeling, data analysis, validation, visualization, writing—original draft. Tatiana Kameneva: Data analysis, visualization, validation, writing—review and editing. Maesoon Im: Validation, writing—review and editing. Shelley I. Fried: Supervision, data analysis, validation, visualization, writing—original draft.

## Competing interests
The authors declare no competing interests.
