## [Peer Review File · Communications Biology]

Reviewers' comments:

Reviewer #1 (Remarks to the Author):

The paper 'Membrane Depolarization Mediates Inhibition ...' by Lee et al. describes the effects of stimulation on retinal ganglion cells. Motivated by clinical application of high-frequency electric stimulation, the focus of the study is to establish effects of electric stimulation across multiple frequencies on neuronal function.

Although the findings are of interest to the field, I have several major concerns.

- Possible generalization of findings on retinal ganglion cells to neuronal networks involved in pain perception or appetite etc. is questionable at best. The retinal network function is significantly different from those in the rest of the nervous system.

Consequently, the finding in this work that the intrinsic effects of the stimuli dominate, may be related to the choice of the retinal preparation.

- Clinical electrostimulation is based on (biphasic) pulse currents. The study here is using sine wave stimulation. Especially, the frequency content of a pulse versus sinusoidal stimulation is different. This seems relevant because this study determines frequency dependent effects of the stimulus.

- I find the paper fairly hard to read, it certainly requires 'work' – largely this is due to the (lack of a) framework in which the story of the results is presented. Stimulus effects are presented as modifications in membrane potential and neuronal firing. In this context I would consider stimulation of a neuron in terms of a nonlinear filter (they show it's a band-filter) where the ion conductivities produce both the nonlinearity as well as the firing properties. I would therefore recommend showing not only horizontal sections across Figures 1D, 3D, 6C,D, but also a few vertical ones (e.g., sections at 40, 70uA in Fig. 3D)

- In the context of the modeling effort I miss the description/background of the bifurcation leading to the depolarization block, a phenomenon that is critical for the findings presented here. The results depicted in e.g. Fig. 7 are a consequence of the bifurcation in this type of dynamics, and the roles of Na and K are given by the (arbitrary) selection of conductivities in the computational model only.

Details

- Considering the focus of the paper, it seems to me that the introduction into retinal aspects could be shorter.

- Unless I missed this, it seems that synaptic function was intact during the experiments. If so, I think that the conclusion that effects are intrinsic is not justified.

- Considering ion channel dynamics (with associated bifurcations), the observed pattern difference between ON and OFF cells is not surprising (as the authors state).

- I don't think the model and experimental results are that similar. There is indeed a qualitative agreement, but the authors overstate the similarity when they write on a few occasions that '... the model closely match the responses observed ...'

- I assume the data was sampled at 10,000 samples/s/channel, correct?

- I have doubt that removing the sinusoidal wave only involves the stimulus artifact, some of the signal may actually be membrane current (e.g., Fig. 2B).

- I don't have sufficient information about the model. Variables are not defined (I can guess, but ...),

voltage sensitive functions are not provided, I would appreciate simulation details. Perhaps an extended description in the supporting material? Perhaps I missed this, but is the simulation software available?

- It might help to mention again in the figure legends why there is a grey or white box top-right in Fig. 1D, 3D.
- Why is there increased firing upon hyperpolarization (Fig. 3A)?
- The change in V_m in the model is not a clear a depolarization as in the recording (Compare Figs. 6A and 2C).
- Fig. 6D, is the color scale in mV units?
- Across multiple figures: what is the unit for spike count (e.g. spikes/s, number of spikes in the response, ...)?
- I don't understand the rationale for Supplementary-Figure 1. Why only the low frequencies. At higher frequencies above $1/(\text{spike duration})$ (about 500Hz) this won't work anymore. Similar to cochlear neuronal responses upon auditory stimuli across a frequency range, higher frequencies will have to elicit different patterns.

Wim van Drongelen
Professor
Section Pediatric Neurology
Computational Neuroscience
The University of Chicago

Reviewer #2 (Remarks to the Author):

The authors investigated the mechanisms of high frequency stimulation (HFS) by directly measuring the membrane potential and spiking activity for two types of retinal ganglion cells (RGCs). Understanding the mechanisms behind selective activation or inhibition of individual cells by HFS can be useful for developing the effective treatment of neurological diseases. They used the whole-cell patch clamp recording to measure the membrane potential and spiking activity of ON and OFF type of alpha-sustained RGCs in explanted mouse retina when stimulation with wide range of frequency (100-2500 Hz) and amplitude (10-100 μA) for duration of 1 second, is applied.

They found that HFS can shift the baseline membrane potential to levels of more depolarization or hyperpolarization which is a function of frequency and amplitude of stimulation. They observed a non monotonic response for HFS (>1KHZ) consistent with previous reports and the baseline voltage became more depolarized, leading to suppression of spiking (depolarization block) which in accordance with some previous modeling studies and rejects the others that claim strong hyperpolarizing currents lead to suppression of spiking.

For intermediate frequency of stimulation (500 Hz), nonmonotonic response is followed by monotonic response and saturated as the amplitude of stimulation is increased. They compared ON and OFF RGCs and showed ON cells were more depolarized than OFF cells when HFS with a given amplitude is applied. The authors also simulated the HFS with a single compartment model neuron by NEURON, and showed that the ion channel dynamics of Na and K account for the shift in the baseline membrane voltage. They showed in the simulation with HFS, the inability of the inactivation gate of sodium (h) to de-inactivate, is

responsible for suppressing spiking activity in the depolarization block. (Regarding HH model neuron, these facts are already known).

Here are some concerns that need to be clarified.

[specific comments]

- 1) The authors need to write explicitly the novelty of the work in the abstract (especially in the last lines).
- 2) How does the spike count change, after the stimulation is finished, say in 1-3 secs after that? For the case of inhibition of neural activity (membrane depolarization) by HFS, does the baseline activity increase (both in experiment and simulation) when the HFS is finished, compared with no stimulation cases?
- 3) When the authors talk about variability in response, it is expected to use some quantitative measure of the variability in spike time or spike count, like coefficient of variation (CV) and/or Fano factor (FF). It is beneficial to make some figures like Fig.1D/Fig.3D to show how variability in experiments (ON and OFF RGC) as well as in their simulations would change as a function of frequency and amplitude of stimulation.
- 4) What are the intrinsic properties of on and off RGCs in this experiment, like baseline firing rate, threshold voltage for firing (V_{θ}), resting voltage (V_r), and membrane time constant for integration of inputs?
Since these parameters may differ from neuron to neuron, for Fig.1, Fig.3 and ...it is better to present the spike count as a function of dimensionless parameter of stimulus amplitude: for example dividing the stimulus amplitude (SA) by $(V_{\theta}-V_r)$ and some time constant for channels or membrane (e.g. $SA/(\tau(V_{\theta}-V_r))$). Similar approach can be taken for membrane potential changes (ΔV_m) with the dimensionless parameter: $\Delta V_m/(V_{\theta}-V_r)$. This new scaling produces a more universal (less neuron dependent) understanding. It helps to compare the result of different neurons that have different intrinsic properties.
- 5) It would be informative to explicitly describe how the spike count is measured. I guess the authors count the number of spikes during the whole stimulation (i.e. 1 s) but this is not explicitly mentioned, at all.
- 6) What are the parameters for gating variables (m,n,h) of Na and K channels for ON and OFF RGCs in the simulation?
- 7) Fig. 8D, three different frequencies (symbols) cannot be distinguished.
- 8) Some abbreviations (PNS, ALS, HCN) were not explained for the first time or at all.
- 9) The manuscript doesn't have any page numbering!

Reviewer #3 (Remarks to the Author):

The manuscript investigates the mechanisms underlying response of retinal ganglion cells (RGCs) to high frequency electrical stimulation using in vitro experiments and computer simulations. The manuscript is technically sound, the methodology is appropriate, and the results are clearly communicated. The authors found a nice way to present the results with different parameters graphically. This study will be of interest to researchers and clinicians working in the areas of pain management, visual prostheses and other fields that require the inhibition or selective stimulation of different neuronal cell types.

I have several suggestions that may help to improve the manuscript.

The health of the cells after continuous high frequency stimulation (HFS) is not clear. How was the health of RGCs measured after stimulation was applied? For how long did you apply stimulation? Did the cells respond in the same fashion at the end of the stimulation? Have you observed adaptation to stimulation? These questions have direct relevance to the clinical translation of the proposed HFS technique.

The cell's response shown in Figure 1B is unusual. Do you have a hypothesis why the second increase in spiking occurs? Does this relate to V_m "second" depolarisation? What are the modelling results in this case?

How quickly V_m return to the resting membrane potential level after the end of the stimulation?

Please clarify if Figures 1 and 3 show the same cell.

Please plot a figure for 500Hz stimulation, in a similar fashion as shown in Figure 5A for 2000 Hz stimulation. Do you observe different responses between ON and OFF cells?

Figure 5B indicates that spike count at peak is not significantly different between On and OFF cells. How does this can be translated into a clinic for selective stimulation?

Why was sensitivity derived in terms of ΔV_m instead of a spike rate?

Very large fluctuations are observed in simulations (Figure 7A1). How can these be physiologically relevant?

Is there a way to measure sodium and potassium currents in response to HFS experimentally?

At what temperature were the experiments performed? Results may change at physiological temperature.

Please show modelling results that replicated experimental data in Figure 5A. I suspect the results will depend not only on sodium and potassium currents and membrane depolarisation, but also on the low-voltage-activated current present in OFF cells.

**Title: Membrane depolarization mediates both the inhibition of neural**
**activity and cell-type-differences in response to high-frequency stimulation.**

**Authors:** Jae-Ik Lee¹, Paul Werginz^{1,2}, Tatiana Kameneva^{3,4}, Maesoon Im^{5,6},

Shelley I. Fried^{1,7*}

**Affiliations:**

¹Department of Neurosurgery, Massachusetts General Hospital, Harvard Medical School;
Boston, MA, 02114, United States of America

²Institute of Biomedical Electronics, TU Wien, 1040 Vienna, Austria

³School of Science, Computing, and Engineering Technologies, Swinburne University of
Technology, Hawthorn, VIC 3122, Australia

⁴Department of Biomedical Engineering, University of Melbourne, Parkville, VIC 3010,
Australia

⁵Brain Science Institute, Korea Institute of Science and Technology (KIST), Seoul, 02792, South
Korea

⁶Division of Bio-Medical Science & Technology, KIST School, University of Science and
Technology (UST), Seoul, 02792, South Korea

⁷Boston VA Healthcare System, Rehabilitation, Research and Development, Boston, MA 02130,
United States of America

*Corresponding author: Shelley I. Fried, 50 Blossom Street, Boston, MA 02114

**ABSTRACT**

Neuromodulation using high frequency (>1 kHz) electric stimulation (HFS) enables
preferential activation or inhibition of individual neural types, thus offering a powerful level of
control that opens the possibility of more effective treatments for a wide range of neurological
diseases. To improve effectiveness, it is important to better understand the mechanisms
governing activation and inhibition with HFS so that selectivity can be optimized. Unfortunately,
it is difficult to directly measure the neurophysiological response to HFS and some previous
modeling studies provide conflicting results. Here, we investigated the mechanisms underlying
HFS-induced inhibition by directly measuring the membrane potential and spiking responses of
ON and OFF types of α -sustained retinal ganglion cells (RGCs) to a wide range of stimulus
frequencies (100 – 2500 Hz) and amplitudes (10 – 100 μ A). We found that HFS leads to shifts in
baseline membrane potential (V_m) and that both the strength as well as the polarity of the shift
were dependent upon the frequency and amplitude of stimulation. Spiking responses in each cell
were directly correlated to the strength and polarity of the shifts in V_m with strong levels of
depolarization leading to spiking suppression. Comparisons between the two cell types revealed
that ON cells were more depolarized by a given amplitude of HFS (vs. OFF cells) – this
sensitivity difference enables the selective targeting. Computational modeling revealed that ion-
channel dynamics largely account for the shifts in baseline membrane potential, suggesting that
an improved understanding of the differences in ion-channel properties across cell types may
enable improved selectivity and ultimately, enhancement of HFS-based strategies for
neurostimulation. In conclusion, our study directly demonstrated that HFS-induced
depolarization led to the suppression of neural activity, and variations in block thresholds and
depolarization sensitivity across different cell types enables preferential activation. Our results

suggest that an improved understanding of the differences in ion-channel properties across cell
types may enable improved selectivity and ultimately, enhancement of HFS-based strategies for
neurostimulation.

INTRODUCTION

High-frequency electric stimulation (HFS) enables novel forms of neuromodulation to be
realized, e.g., blocking neural activity¹⁻⁴ or preferentially activating (or inhibiting) specific types of
fibers⁵⁻⁸ or neurons^{9,10}. Clinical studies that harness some of the innovative features of HFS have
demonstrated promising results, including significant and sustained pain reduction in patients with
postamputation¹¹ or chronic back/lower limb pain¹², as well as weight loss in obese individuals¹³. Unlike
pharmacological agents, the effectiveness of HFS is almost instantaneous^{4, 14-16}, with no evidence of
deterioration of residual function¹¹, and it has not been associated with significant adverse effects even
after chronic use^{17, 18}.

Intriguingly, individual types of nerves or nerve fibers within a heterogeneous population can
exhibit different sensitivities to HFS, e.g., nerve fibers that carry pain signals can be blocked while fibers
that carry other sensations or mediate motor activity are not disrupted^{19,20}. Such selectivity is especially
attractive for applications in which a large diversity of neural types are present, e.g., in the retina, upwards
of 15 different types of ganglion cells (RGCs, retinal output neurons)^{21,22} each elicit distinct patterns of
spiking in response to a given light stimulus²³⁻²⁵. It is challenging, however, to create distinct patterns of
spiking in adjacent neurons with a prosthesis and the transmission of non-physiological signals to the
brain likely contributes to the sub-optimal results obtained to date. The ability to selectively target
individual types with a prosthesis offers the potential to reproduce important elements of natural signaling
and therefore improve clinical outcomes. As one example, ON and OFF types of retinal ganglion cells
(RGCs) have similar sensitivity to individual biphasic pulses^{26,27} and thus the out-of-phase spiking that
occurs naturally in these two types is difficult to replicate with conventional stimulation techniques. High-
density, high-count arrays may enable some selectivity [Ref], but these require very close proximity to the
targeted neurons, which may be difficult to reliably sustain in clinical applications. Such an approach may
also have limited utility in the fovea where RGC somas can be stacked 6 or 7 high.
[revised manuscript text omitted]

capture both spiking activity and V_m in the same cell revealed that the two were correlated with
suppression of spiking occurring only for strong levels of depolarization, thereby providing strong
support for the notion that HFS-induced inhibition of spiking arises from depolarization block. In general,
any combination of stimulus parameters that caused the depolarizing change in V_m to exceed a certain
threshold level-initiated depolarization block. Results from our computational model demonstrated a
qualitative agreement with physiological experiments and revealed that prolonged inactivation of Na_v
channels during continued depolarization (depolarization block) contributes to the cessation of spiking
(Figure 9). Interestingly, the model consisted of only a single compartment and therefore, the similarity to
physiological results suggests the intrinsic dynamics of voltage-gated ion channels are largely responsible
for the biphasic shifts in V_m i.e., the observed responses to HFS are not significantly influenced by
morphological features of RGCs, or synaptic circuitry.

Note that the responses to HFS using sinusoidal waveforms are highly similar to previous studies
using HFS using pulsatile waveforms^{9,10}. This is consistent with previous studies outside the retina which
showed that responses to high frequency stimulation was similar for the two different waveforms^{2,63,64}.
For example, similar to previous studies, HFS-induced inhibition occurred primarily at frequencies above
$\sim 1 \text{ kHz}^2$. Also consistent with previous studies, the relationship between the threshold current for
suppressing neural activity and stimulus frequency was linear (Figure 4E)^{2,63,64}. The reduced
depolarization for higher stimulation frequencies likely occurs because a smaller amount of electric
charge was delivered per pulse and a shorter duration of the depolarizing phase caused Na_v channels to be
less active, necessitating greater stimulus current for the cells to reach a given level of depolarization.
This is consistent with our modeling results which indicate that at high rates, inward currents through
relatively fast-acting sodium (Na_v) channels predominate over outward currents through slower potassium
(K_v) channels (Figure 7), leading to depolarization and eventually depolarization block. The similarity of
our findings to previous work supports the notion that depolarization block is also the mechanism
mediating suppressed activation in other HFS studies and clinical settings, although it is important to note
that variations in methodology, e.g., stimulus waveform, the neural system being studied (PNS vs retina)
and/or animal species, may all influence the response to HFS and therefore, mechanisms other than
depolarization block may also contribute to HFS-induced inhibition in other studies.

Furthermore, our observations of unusual spiking activity with a second increase at high stimulus
amplitudes (Figure 1B) raise interesting questions regarding the interplay between membrane potential
dynamics and the activation of sodium and potassium channels. While our study primarily focuses on the
suppression of neural activity by high-frequency stimulation, future investigations should delve deeper
into the mechanisms underlying these intriguing response patterns and their potential implications.

**What underlies the variation in responses across cell types?**

Interestingly, the level of V_m at which depolarization block occurred was different for ON vs.
OFF types of alpha sustained RGCs: depolarization block occurred for a ΔV_m of 15.89 ± 3.86 mV in ON
cells vs. 11.24 ± 3.20 mV for OFF cells (Figure 5D and E). The more depolarized level in ON cells was
somewhat surprising since their peak firing rates tended to occur at lower stimulus amplitudes than in
OFF cells (Figure 5A and C). To better understand this apparent discrepancy, we compared the level of
depolarization that arose in ON vs. OFF cells as a function of the amplitude of HFS and found that a
given increase in amplitude produced a larger depolarization in OFF (vs. ON) cells (Figure 5F). Thus,
even though depolarization block does not occur in ON cells until the cell is more depolarized, the
sensitivity of ΔV_m to stimulus strength is higher in ON cells and thus the stimulus amplitude at which the
onset of depolarization block occurs is lower in ON vs. OFF cells.

Our results do not reveal the reasons for the sensitivity differences between ON and OFF alpha
RGCs (either V_m at threshold or the change in V_m as a function of stimulus strength), although they do
suggest that both arise from differences in the intrinsic properties of the two cell types. In the PNS,
responses to HFS are necessarily mediated by the kinetics of the voltage-sensitive ion channels in the
axonal membrane given that the somas are too distant to be activated by the stimulus and there are no
synaptic connections elsewhere along the axon. In the retina, responses to HFS are likewise mediated
intrinsic to the cell given that they are not sensitive to blockers of synaptic input^{9, 10, 52, 55}. Therefore,
although we did not repeat the control experiment blocking synaptic input in this study, it is likely that
RGC responses to HFS are similarly mediated by voltage-sensitive ion channels within the cell
membrane. If responses to HFS do indeed arise intrinsically, then response differences between ON and
OFF RGCs are likely to arise from differences in channel types, expression levels and/or kinetics of the
voltage-sensitive ion channels within the cell membrane. Consistent with this, there are known
differences in the types of voltage-gated calcium channels expressed in ON vs. OFF types of alpha cells⁶⁵
and Boal *et al.*⁶⁶ reported that OFF α sustained RGCs are more sensitive to extracellular potassium
concentration than ON α sustained RGCs, suggesting an intrinsic difference in potassium channel

properties between the same two types. Further support comes from a previous HFS modeling study
which demonstrated that increasing the conductance of voltage-gated potassium channels resulted in a
shift of the peak responses to stronger amplitudes of HFS³⁷. Along similar lines, Guo *et al.*³⁸ suggested
that increasing the concentration of sodium in the extracellular solution resulted in an increase in the
stimulus amplitude at which spiking responses peaked, suggesting differences in sodium channel
properties between ON and OFF types.

Voltage-gated sodium and potassium channels are densely packed in the axon initial segment
(AIS), which is known to play a critical role in the initiation and regulation of spike generation⁶⁷. Werginz
*et al.*⁵⁹ showed that, within the OFF α transient subtype, RGCs with longer AISs enter depolarization
block at more depolarized levels of V_m in response to somatic injections of depolarizing current. While
the specific components within the AIS that determine the level at which depolarization is reached have
not been identified, the AISs of ON α sustained RGCs are longer than those of nearby OFF α sustained
cells³⁹, perhaps accounting for the higher V_m thresholds identified here (Figure 5E) (but see Boal *et al.*⁶⁶).
Consistent with this, cells with longer AISs had peak responses at weaker stimulus amplitudes, i.e.,
entered depolarization block at more depolarized levels of V_m ³⁷. In future studies, it will be interesting to
further characterize the intrinsic anatomical differences across types and then determine which ones can
be harnessed to improve the selectivity of targeting with HFS, both in the retina as well as other regions
of the central/peripheral nervous systems.

[revised manuscript text omitted]

**Competing interests**

The authors have no competing interests to declare.

**Data and Code availability**

The source data and codes are available on the Open Science Framework

(<https://doi.org/10.17605/osf.io/hnpq4>).

**Author contributions**

Jae-Ik Lee: Conceptualization, Physiological experiment, Data analysis, Methodology, Visualization,

Writing – original draft.

Paul Werginz: Computational modeling, Data analysis, Validation, Visualization, Writing – original draft.

Tatiana Kameneva: Data analysis, Visualization, Validation, Writing – review and editing.

Maesoon Im: Validation, Writing – review and editing.

Shelley I. Fried: Supervision, Data analysis, Validation, Visualization, Writing – original draft.

**REFERENCES**

- 1. Cai, C., Ren, Q., Desai, N.J., Rizzo, J.F., 3rd & Fried, S.I. Response variability to high rates of
electric stimulation in retinal ganglion cells. *J Neurophysiol* **106**, 153-162 (2011).
- 2. Kilgore, K.L. & Bhadra, N. Reversible nerve conduction block using kilohertz frequency
alternating current. *Neuromodulation* **17**, 242-254; discussion 254-245 (2014).
- 3. Bhadra, N., Vrabec, T.L., Bhadra, N. & Kilgore, K.L. Reversible conduction block in peripheral
nerve using electrical waveforms. *Bioelectron Med (Lond)* **1**, 39-54 (2018).
- 4. Patel, Y.A. & Butera, R.J. Challenges associated with nerve conduction block using kilohertz
electrical stimulation. *J Neural Eng* **15**, 031002 (2018).
- 5. Patel, Y.A. & Butera, R.J. Differential fiber-specific block of nerve conduction in mammalian
peripheral nerves using kilohertz electrical stimulation. *J Neurophysiol* **113**, 3923-3929 (2015).
- 6. Joseph, L. & Butera, R.J. Unmyelinated Aplysia nerves exhibit a nonmonotonic blocking response
to high-frequency stimulation. *IEEE Trans Neural Syst Rehabil Eng* **17**, 537-544 (2009).
- 7. Joseph, L. & Butera, R.J. High-frequency stimulation selectively blocks different types of fibers in
frog sciatic nerve. *IEEE Trans Neural Syst Rehabil Eng* **19**, 550-557 (2011).
- 8. Chang, Y.C. et al. kHz-frequency electrical stimulation selectively activates small, unmyelinated
vagus afferents. *Brain Stimul* **15**, 1389-1404 (2022).
- 9. Twyford, P., Cai, C. & Fried, S. Differential responses to high-frequency electrical stimulation in
ON and OFF retinal ganglion cells. *J Neural Eng* **11**, 025001 (2014).
- 10. Muralidharan, M. et al. Neural activity of functionally different retinal ganglion cells can be
robustly modulated by high-rate electrical pulse trains. *J Neural Eng* **17**, 045013 (2020).
- 11. Sooin, A., Shah, N.S. & Fang, Z.P. High-frequency electrical nerve block for postamputation pain: a
pilot study. *Neuromodulation* **18**, 197-205; discussion 205-196 (2015).
- 12. Kapural, L. et al. Comparison of 10-kHz High-Frequency and Traditional Low-Frequency Spinal
Cord Stimulation for the Treatment of Chronic Back and Leg Pain: 24-Month Results From a
Multicenter, Randomized, Controlled Pivotal Trial. *Neurosurgery* **79**, 667-677 (2016).
- 13. Apovian, C.M. et al. Two-Year Outcomes of Vagal Nerve Blocking (vBloc) for the Treatment of
Obesity in the ReCharge Trial. *Obes Surg* **27**, 169-176 (2017).
- 14. Bhadra, N. & Kilgore, K.L. High-frequency electrical conduction block of mammalian peripheral
motor nerve. *Muscle Nerve* **32**, 782-790 (2005).
- 15. Williamson, R.P. & Andrews, B.J. Localized electrical nerve blocking. *IEEE Trans Biomed Eng* **52**,
362-370 (2005).
- 16. Pelot, N.A. & Grill, W.M. In vivo quantification of excitation and kilohertz frequency block of the
rat vagus nerve. *J Neural Eng* **17**, 026005 (2020).
- 17. Gaunt, R.A. & Prochazka, A. Transcutaneously coupled, high-frequency electrical stimulation of
the pudendal nerve blocks external urethral sphincter contractions. *Neurorehabil Neural Repair*
**23**, 615-626 (2009).
- 18. Rubinstein, J.T., Tyler, R.S., Johnson, A. & Brown, C.J. Electrical suppression of tinnitus with high-
rate pulse trains. *Otol Neurotol* **24**, 478-485 (2003).
- 19. Zhao, S. et al. Conduction block in myelinated axons induced by high-frequency (kHz) non-
symmetric biphasic stimulation. *Front Comput Neurosci* **9**, 86 (2015).
- 20. Rapeaux, A., Nikolic, K., Williams, I., Eftekhari, A. & Constandinou, T.G. Fiber size-selective
stimulation using action potential filtering for a peripheral nerve interface: A simulation study.
*Annu Int Conf IEEE Eng Med Biol Soc* **2015**, 3411-3414 (2015).
- 21. Sanes, J.R. & Masland, R.H. The types of retinal ganglion cells: current status and implications for
neuronal classification. *Annu Rev Neurosci* **38**, 221-246 (2015).

- 22. Baden, T. et al. The functional diversity of retinal ganglion cells in the mouse. *Nature* **529**, 345-
350 (2016).
- 23. O'Brien, B.J., Isayama, T., Richardson, R. & Berson, D.M. Intrinsic physiological properties of cat
retinal ganglion cells. *J Physiol* **538**, 787-802 (2002).
- 24. Meister, M. & Berry, M.J., 2nd The neural code of the retina. *Neuron* **22**, 435-450 (1999).
- 25. Schnitzer, M.J. & Meister, M. Multineuronal firing patterns in the signal from eye to brain.
*Neuron* **37**, 499-511 (2003).
- 26. Fried, S.I., Hsueh, H.A. & Werblin, F.S. A method for generating precise temporal patterns of
retinal spiking using prosthetic stimulation. *J Neurophysiol* **95**, 970-978 (2006).
- 27. Sekirnjak, C. et al. High-resolution electrical stimulation of primate retina for epiretinal implant
design. *J Neurosci* **28**, 4446-4456 (2008).
- 28. Jepson, L.H. et al. Spatially patterned electrical stimulation to enhance resolution of retinal
prostheses. *J Neurosci* **34**, 4871-4881 (2014).
- 29. Jepson, L.H. et al. Focal electrical stimulation of major ganglion cell types in the primate retina
for the design of visual prostheses. *J Neurosci* **33**, 7194-7205 (2013).
- 30. Grosberg, L.E. et al. Activation of ganglion cells and axon bundles using epiretinal electrical
stimulation. *J Neurophysiol* **118**, 1457-1471 (2017).
- 31. Madugula, S.S. et al. Focal electrical stimulation of human retinal ganglion cells for vision
restoration. *J Neural Eng* **19** (2022).
- 32. Tai, C., de Groat, W.C. & Roppolo, J.R. Simulation of nerve block by high-frequency sinusoidal
electrical current based on the Hodgkin-Huxley model. *IEEE Trans Neural Syst Rehabil Eng* **13**,
415-422 (2005).
- 33. Tai, C., Wang, J., Roppolo, J.R. & de Groat, W.C. Relationship between temperature and
stimulation frequency in conduction block of amphibian myelinated axon. *Journal of*
*computational neuroscience* **26**, 331-338 (2009).
- 34. Zhang, X., Roppolo, J., de Groat, W. & Tai, C. Simulation analysis of nerve block by high
frequency biphasic electrical current based on frankenhaeuser-huxley model. *Conf Proc IEEE Eng*
*Med Biol Soc* **2005**, 4247-4250 (2005).
- 35. Zhang, X., Roppolo, J.R., de Groat, W.C. & Tai, C. Mechanism of nerve conduction block induced
by high-frequency biphasic electrical currents. *IEEE Trans Biomed Eng* **53**, 2445-2454 (2006).
- 36. Ackermann, D.M., Bhadra, N., Gerges, M. & Thomas, P.J. Dynamics and sensitivity analysis of
high-frequency conduction block. *J Neural Eng* **8**, 065007 (2011).
- 37. Kameneva, T. et al. Retinal ganglion cells: mechanisms underlying depolarization block and
differential responses to high frequency electrical stimulation of ON and OFF cells. *J Neural Eng*
**13**, 016017 (2016).
- 38. Guo, T. et al. Mediating Retinal Ganglion Cell Spike Rates Using High-Frequency Electrical
Stimulation. *Front Neurosci* **13**, 413 (2019).
- 39. Raghuram, V., Werginz, P. & Fried, S.I. Scaling of the AIS and Somatodendritic Compartments in
alpha S RGCs. *Front Cell Neurosci* **13**, 436 (2019).
- 40. Pyragas, K., Novičenko, V. & Tass, P.A. Mechanism of suppression of sustained neuronal spiking
under high-frequency stimulation. *Biological cybernetics* **107**, 669-684 (2013).
- 41. Wilson, D. Stabilization of weakly unstable fixed points as a common dynamical mechanism of
high-frequency electrical stimulation. *Sci. Rep.* **10**, 5922 (2020).
- 42. Weinberg, S.H. High-frequency stimulation of excitable cells and networks. *PLoS One* **8**, e81402
(2013).
- 43. Wang, H., Wang, S., Gu, Y. & Yu, Y. Hopf Bifurcation Analysis of a Two-Dimensional Simplified
Hodgkin–Huxley Model. *Mathematics* **11**, 717 (2023).

- 44. Amarillo, Y., Mato, G. & Nadal, M.S. Analysis of the role of the low threshold currents IT and Ih in
intrinsic delta oscillations of thalamocortical neurons. *Frontiers in computational neuroscience* **9**,
52 (2015).
- 45. Erhardt, A.H. Bifurcation analysis of a certain Hodgkin-Huxley model depending on multiple
bifurcation parameters. *Mathematics* **6**, 103 (2018).
- 46. Qian, K., Yu, N., Tucker, K.R., Levitan, E.S. & Canavier, C.C. Mathematical analysis of
depolarization block mediated by slow inactivation of fast sodium channels in midbrain
dopamine neurons. *J. Neurophysiol.* **112**, 2779-2790 (2014).
- 47. Dovzhenok, A. & Kuznetsov, A.S. Exploring neuronal bistability at the depolarization block.
(2012).
- 48. Izhikevich, E.M. Neural excitability, spiking and bursting. *International journal of bifurcation and*
*chaos* **10**, 1171-1266 (2000).
- 49. Bose, A. (2014).
- 50. Cowan, J., Neuman, J., Kiewiet, B. & Van Drongelen, W. Self-organized criticality in a network of
interacting neurons. *Journal of statistical mechanics: theory and experiment* **2013**, P04030
(2013).
- 51. Van Drongelen, W. Modeling neural activity. *International Scholarly Research Notices* **2013**
(2013).
- 52. Freeman, D.K., Eddington, D.K., Rizzo, J.F., 3rd & Fried, S.I. Selective activation of neuronal
targets with sinusoidal electric stimulation. *J Neurophysiol* **104**, 2778-2791 (2010).
- 53. Lee, J.I. & Im, M. Optimal Electric Stimulus Amplitude Improves the Selectivity Between
Responses of ON Versus OFF Types of Retinal Ganglion Cells. *IEEE Trans Neural Syst Rehabil Eng*
**27**, 2015-2024 (2019).
- 54. Werginz, P., Im, M., Hadjinicolaou, A.E. & Fried, S.I. Visual and electric spiking responses of
seven types of rabbit retinal ganglion cells. *Annu Int Conf IEEE Eng Med Biol Soc* **2018**, 2434-
2437 (2018).
- 55. Cai, C., Twyford, P. & Fried, S. The response of retinal neurons to high-frequency stimulation. *J*
*Neural Eng* **10**, 036009 (2013).
- 56. Tsai, D., Morley, J.W., Suaning, G.J. & Lovell, N.H. Direct activation and temporal response
properties of rabbit retinal ganglion cells following subretinal stimulation. *J Neurophysiol* **102**,
2982-2993 (2009).
- 57. Bleckert, A., Schwartz, G.W., Turner, M.H., Rieke, F. & Wong, R.O. Visual space is represented by
nonmatching topographies of distinct mouse retinal ganglion cell types. *Current Biology* **24**, 310-
315 (2014).
- 58. Warwick, R.A., Kaushansky, N., Sarid, N., Golan, A. & Rivlin-Etzion, M. Inhomogeneous Encoding
of the Visual Field in the Mouse Retina. *Curr Biol* **28**, 655-665 e653 (2018).
- 59. Werginz, P., Raghuram, V. & Fried, S.I. Tailoring of the axon initial segment shapes the
conversion of synaptic inputs into spiking output in OFF-alpha T retinal ganglion cells. *Sci Adv* **6**,
37 (2020).
- 60. Fohlmeister, J.F., Cohen, E.D. & Newman, E.A. Mechanisms and distribution of ion channels in
retinal ganglion cells: using temperature as an independent variable. *J. Neurophysiol.* **103**, 1357-
1374 (2010).
- 61. Craven, K.B. & Zagotta, W.N. CNG and HCN channels: two peas, one pod. *Annu Rev Physiol* **68**,
375-401 (2006).
- 62. Wahl-Schott, C. & Biel, M. HCN channels: structure, cellular regulation and physiological
function. *Cell Mol Life Sci* **66**, 470-494 (2009).
- 63. Peña, E., Pelot, N.A. & Grill, W.M. Quantitative comparisons of block thresholds and onset
responses for charge-balanced kilohertz frequency waveforms. *J. Neural Eng.* **17**, 046048 (2020).

- 64. Peña, E., Pelot, N.A. & Grill, W.M. Non-monotonic kilohertz frequency neural block thresholds
arise from amplitude- and frequency-dependent charge imbalance. *Sci Rep* **11**, 5077 (2021).
65. Margolis, D.J., Gartland, A.J., Euler, T. & Detwiler, P.B. Dendritic calcium signaling in ON and OFF
mouse retinal ganglion cells. *J Neurosci* **30**, 7127-7138 (2010).
66. Boal, A.M., McGrady, N.R., Risner, M.L. & Calkins, D.J. Sensitivity to extracellular potassium
underlies type-intrinsic differences in retinal ganglion cell excitability. *Front Cell Neurosci* **16**,
966425 (2022).
67. Bender, K.J. & Trussell, L.O. The physiology of the axon initial segment. *Annu Rev Neurosci* **35**,
249-265 (2012).
68. Pang, J.J., Gao, F. & Wu, S.M. Light-evoked excitatory and inhibitory synaptic inputs to ON and
OFF alpha ganglion cells in the mouse retina. *J Neurosci* **23**, 6063-6073 (2003).
69. Murphy, G.J. & Rieke, F. Network variability limits stimulus-evoked spike timing precision in
retinal ganglion cells. *Neuron* **52**, 511-524 (2006).
70. Barlow, H.B., Hill, R.M. & Levick, W.R. Retinal Ganglion Cells Responding Selectively to Direction
and Speed of Image Motion in the Rabbit. *J Physiol* **173**, 377-407 (1964).
71. Hines, M.L. & Carnevale, N.T. The NEURON simulation environment. *Neural Comput* **9**, 1179-
1209 (1997).

**Figure 1. RGC response patterns show variation with stimulus frequency.**

(A-C) The number of spikes elicited in an ON α sustained RGC is plotted as a function of current
 amplitude for stimulus frequencies of 200 (A), 500 (B), and 1,500 Hz (C). Arrowheads indicate the
 amplitude level at which monotonic curves reached the maximum response, and asterisks indicate the
 amplitude at which non-monotonic responses peaked. (D) The number of spikes elicited in a
 representative cell for all combinations of frequency and amplitude is mapped. Consistent with (A-C),
 arrowheads and asterisks indicate plateau levels and peak responses, respectively. The rows indicated by
 white arrows (between the heat map and the color bar) correspond to the responses shown in (A-C). The
 stimulus amplitudes of 90 – 100 μA for 100 Hz (colored in gray) were not tested to prevent potential
 damage on stimulating electrodes and neurons by excessive charge.

A. Raw recording

B. Stimulus artifact

C. Action potentials + Baseline membrane potential

**Figure 2. Sinusoidal stimulation leads to changes in baseline membrane potential**

(A) A whole-cell recording of a representative RGC response to 2,000 Hz sinusoidal stimulation (duration
of 1 second and amplitude of 50 μ A). An expanded view of the recording at stimulus onset (blue box) is
shown below. (B) Stimulus artifact extracted from the whole-cell recording by fitting with a sinusoidal
function. (C) After removing the stimulus artifact from the raw recording (i.e., A – B), the baseline
membrane potential (red trace) is obtained by median filtering (see Methods). ΔV_m was quantified by the
difference between resting membrane potential (i.e., membrane potential before stimulus onset) and
shifted baseline membrane potential during stimulation. Scale bar in B and C is the same as in A (inset).

**Figure 3. The change in baseline membrane potential varies with stimulus frequency.**

(A-C) The average membrane potential changes (ΔV_m) induced by stimulus frequencies of 200 (A), 500
 (B), and 1500 Hz (C) are plotted as a function of the stimulus amplitude. (D) The heat map shows ΔV_m
 induced by each stimulus condition in a representative cell (the same cell in Figure 1). The positive and
 negative signs in the legend indicate depolarization and hyperpolarization, respectively. The rows
 indicated by white arrows (between the heat map and the color bar) correspond to membrane potential
 changes shown in (A-C).

**Figure 4. Excessive depolarization leads to spike suppression.**

(A-C) Typical whole-cell patch recordings (artifact-subtracted, see Methods) at low, medium, and high
levels of depolarization, respectively. Red traces show the shift in baseline membrane potential. Stimulus
currents are 20 (A), 40 (B), and 60 μ A (C). The dotted blue horizontal lines are arbitrarily positioned to
help facilitate comparisons of spike amplitude. (D-E) The number of spikes is plotted as a function of
ΔV_m (D) and as a function of current amplitude (E) for stimulation frequencies of 1,000, 1,500 and
738 Hz. (F-G) The number of spikes is plotted as a function of ΔV_m (F) and current amplitude (G) in response
to 2,000 Hz stimulation. The black and gray line segments indicate portions of the curve below and above
the peak firing rate, respectively. Error bars are omitted for clarity.

**Figure 5. The differences in the responses of ON vs. OFF RGCs to HFS arises from differences in**
 **their sensitivity to HFS as well as the voltage level at which depolarization block occurs in each**
 **type.**

(A) The number of spikes elicited in ON ($n = 11$) and OFF ($n = 8$) cells as a function of stimulus
 amplitude of 2,000 Hz stimulation. Thin lines indicate the average per cell, and the thick solid lines
 indicate the average per population. Error bars show the standard error of the mean (SEM). (B and C)
 Comparisons of peak spike counts (B) and stimulus amplitudes at which peak responses were elicited,
 I_{peak} , (C) between the ON and OFF cells. (D) The induced membrane potential change, ΔV_m , as a function
 of stimulus amplitude. Each line indicates the average per cell, and the circles indicate the breakdown
 point where the elicited spikes begin to decrease. (E and F) Comparisons of breakdown ΔV_m (E) and
 sensitivity (F) between the ON and OFF cells. The sensitivity was defined by breakdown ΔV_m divided by
 the corresponding stimulus amplitude.

Figure 6. Modeling results closely match the results from physiological experiments.

(A, top) Schematic of the stimulation and recording configuration as well as the equivalent electric circuit of the single-compartment model. Stimulation was applied intracellularly by injecting a sinusoidal waveform with the same frequencies used physiologically. (A, bottom) Representative trace of membrane potential over time in response to high-frequency stimulation. (B) Spiking response is plotted versus the amplitude of the injected current for stimulus rates of 200, 700, and 1,500 Hz. (C-D) Heat maps showing spike counts (C) and ΔV_m (D) for all frequency and amplitude combinations tested. Color bars at right.

**Figure 7. Sodium and potassium channels have different sensitivities to low vs. high-frequency**
 **stimulation.**

(A1) Artifact-subtracted membrane potential (black) and ΔV_m (red) for 200 Hz stimulation. The dotted
 line indicates the resting membrane potential. (A2) Expanded view of the membrane potential in A1,
 corresponding to one period of the sinusoidal waveform. (A3) Sodium (blue) and potassium (orange)
 currents correspond to A2. The x-axis is applied to both A2 and A3. The total charges transferred through
 the sodium (q_{Na}) and potassium (q_K) channels were calculated by the area under each curve. (B1-3) Same
 as A1-A3 but for 2,000 Hz stimulation. B2 and B3 show the traces for the membrane potential and ion
 currents for 5 ms (as A2 and A3), which correspond to ten periods of the sinusoidal waveform. In the
 inset of B3, the logarithmic scale was used on the y-axis to visually accommodate both sodium and
 potassium currents of which peak values differ by more than an order.

**Figure 8. Ionic current imbalance leads to differential membrane polarization during HFS.**

(A) For 200 Hz stimulation, sodium (blue) and potassium (orange), as well as the summed (black) charges
 were plotted as a function of stimulus amplitude. The charge was recorded for a period of 10 ms starting
 500 ms after pulse onset. (B and C) Same as A but for 500 (B) and 2,000 Hz (C). (D) ΔV_m induced by
 each stimulus amplitude of 200, 500, and 2,000 Hz stimulation is plotted as a function of the summed
 charge for (Pearson's r value: 0.99).

**Figure 9. The inactivation of the Na channel underlies the depolarization block.**

(A) Transition into depolarization block during 2,000 Hz stimulation. The black trace indicates artifact-

subtracted membrane potential, and the red line indicates ΔV_m , i.e., baseline membrane potential. (B)

Corresponding sodium m and h gating variables (blue and orange, respectively) are plotted over time.

**Supplementary Figure 1. Responses to low-frequency (100 – 300 Hz) stimulation**

(A) The number of elicited spikes by 100 – 300 Hz stimulation is plotted as a function of stimulus
 amplitude. (B) Action potentials (black traces) and stimulus artifacts (blue traces) extracted from raw
 recordings. Stimulus amplitude was fixed as 80 μA. Red dots on the stimulus artifacts (i.e., blue traces)
 indicate the timing of the onset of action potentials.

**Supplementary Figure 2. Spiking response patterns are correlated with ΔV_m**

Combinations of stimulus amplitudes and frequencies that hyperpolarized the membrane potential colored
 red, while those depolarized the membrane potential colored blue. For each frequency, the range of
 stimulus amplitudes that led to monotonic responses was indicated by a two-headed blue arrow, while the
 amplitudes that led to non-monotonic responses were indicated by a red arrow.

**Supplementary Figure 3. Spiking activity and ΔV_m in response to the change in stimulus frequency**

(A and B) The number of spikes (red lines) and the baseline membrane potential changes (ΔV_m , blue
lines) in an ON α sustained RGC (the same cell as in Fig. 1 and 3) are plotted as a function of stimulus
frequency. The current amplitudes were held constant at 40 (A) and 70 μA (B). The error bars indicate
standard error.

**Supplementary Figure 4. An example of bifurcation observed in experimental data.**

A representative RGC response to 2,000 Hz stimulation. In this case, the bifurcation parameter is
 stimulus amplitude and state of the dynamical system is represented as membrane potential. Blue, red:
 maximum and minimum of the membrane potential, respectively. Three operating regimes are shown for
 this neuron: 1) stable no spiking region, 2) repetitive firing region, and 3) no spiking region for high
 amplitude stimulation. In contrast to the simplified mathematical model bifurcation plots, this figure has
 two lines for maximum and minimum of the membrane potential in regions 1 and 2 due to fluctuations of
 the membrane potential in non-spiking regimes.

 **Supplementary Figure 5. A representative recording showing the post-stimulus change in the**
 **membrane potential**
 A representative recording of RGC response to 2,000 Hz stimulation (black). The red and blue lines
 indicate the baseline membrane potential and resting membrane potential level, respectively. Strong
 HFS resulting in a substantial depolarization was followed by a momentary hyperpolarization below the
 resting level. This post-stimulus change in the membrane potential typically lasted for less than 500 ms.

 **Supplementary Figure 6. The differences in the responses of ON vs. OFF RGCs to 500 Hz**
 **stimulation**
 (A) The number of spikes elicited in ON ($n = 4$) and OFF ($n = 4$) cells as a function of stimulus amplitude
 of 500 Hz stimulation. The asterisks indicate the peak spike counts of non-monotonic response part, and
 the circles indicate the transition point from non-monotonic to monotonic responses. (B) The induced
 membrane potential change, ΔV_m , as a function of stimulus amplitude. The asterisks indicate the peak
 depolarization levels.

Reviewers' comments:

Reviewer #1 (Remarks to the Author):

The revised manuscript is improved. As I outline below (I use the numbering the authors used in their rebuttal), several problems were insufficiently addressed. The outline includes, at the Editor's request, a number of points brought up by Reviewer#3 (see items labeled with, Rev#3)

#2) and #7)

- I find the motivation in the rebuttal is satisfactory (although it's a bit of an excuse why the neurons were not synaptically isolated while this would have been easy to do). I would prefer this motivation to be more clearly stated and discussed in the manuscript.

- There are other studies that show different depolarization block properties across different types of neurons (e.g., threshold differences in excitatory vs. inhibitory cells in neocortex and hippocampus). #4)

- The readability is improved but addition of essential graphs in the supplement (e.g., Suppl. Fig. 3), rather than include them as a panel in an existing figure, doesn't necessarily help in my opinion.

- Rev#3: The above comment is also relevant for Suppl. Figs. 5 and 6.

#11)

- The model description is improved. I do have an additional reaction; See #11, Rev#3 below

#14)

- I do not understand the reply and how this was addressed in the revision.

#6, Rev#3)

- It is not clear to me how the reply to point 2 is reflected in the revised text.

#11, Rev#3)

- Bringing up different types of currents can be relevant for the physiological interpretation. However, the modeling results are to be interpreted in light of the limitations of Hodgkin & Huxley type models. Because not all currents are known, certainly not in sufficient detail, parameters must be fudged a bit (in physics one might call this renormalization ... :-). I miss an honest assessment of that limitation.

Reviewer #2 (Remarks to the Author):

1. It is interesting that there is a domain in which on-RGCs are suppressed while off-RGCs respond under HFS which leads to selective targeting of cells. Since this sensitivity difference is mentioned in the title/abstract and is one of the authors main focuses in this paper, it is worth to show computationally the sensitivity differences across two RGC types under HFS by modelling. What factor determines the sensitivity differences in response to HFS in these two types of the cells from computational study?

2. What do authors expect when HFS is applied to different neurons in different regions of the brain with burst spiking, high spontaneous firings (like Purkinje cells)? It is good to show the dependency of the effect on different regimes of spontaneous firing by computational modelling.

3. The authors examined 1s duration for HFS. It is good to discuss how the result (inhibition of neural activity) changes for different duration of stimulation (lower/ higher than 1s)?

4. All the experimental results are taken from explanted mice (which would be better to mention explicitly in the caption of relevant figures). The result can be different for in vivo cells in living animals which need to be discussed.

5. It is better for the reader to know about the error bars (whether standard deviation or SEM) in each figure caption (Figs. 1, 3, 4,...)

6. Figure 6B: Is the frequency for the last graph 1500 Hz or 2000 Hz? (Text in caption and figure are not consistent.)

7. Line 257-258: "monotonic responses for low frequencies of stimulation (Figure 6B, top), non-monotonic responses for high frequencies (Figure 6B, 257 bottom) and combination of both for intermediate frequencies (Figure 6B, middle)."

There is no top/middle/bottom in Figure 6B.

8. "To better understand this apparent discrepancy, we compared the level of depolarization that arose in ON vs. OFF cells as a function of the amplitude of HFS and found that a given increase in amplitude produced a larger depolarization in OFF (vs. ON) cells (Figure 5F)."

A given increase in amplitude produces a larger depolarization in ON cells (vs. OFF).

Reviewer #1

1) [#2 and #7] I find the motivation in the rebuttal is satisfactory (although it's a bit of an excuse why the neurons were not synaptically isolated while this would have been easy to do). I would prefer this motivation to be more clearly stated and discussed in the manuscript. Also, there are other studies that show different depolarization block properties across different types of neurons (e.g., threshold differences in excitatory vs. inhibitory cells in neocortex and hippocampus).

We appreciate the comment and agree that our manuscript did not elaborate adequately on our methodological approach. In response, we have revised the manuscript to incorporate a more thorough explanation of our methodology, including the limitations arising from not using synaptic blockers. We note, as an aside, that while the addition of synaptic blockers is methodologically straightforward, and employed routinely in our lab, the ‘difficulty’ with these experiments arose from time limitations. Specifically, running through the full matrix of all frequency and amplitude combinations took some time, and had to be repeated both for spiking and for membrane voltage (to create the maps in Figures 1F and 3F). There were additional control experiments needed for each cell, e.g., the responses to light stimuli used to determine cell type, and thus, the difficulty arose, in part, from limitations in how long we could reliably hold the cell via whole cell patch clamp.

We appreciate the information on studies that show depolarization block properties in different types of neurons. Even though it is outside the field of high-frequency stimulation, we agree that such sensitivity differences are relevant to our findings here and have mentioned the finding in the revised text.

Line 512:

Synaptic inputs were not blocked pharmacologically during our investigation of RGC responses to HFS. While blocking such input is well established for RGCs, such experiments would have necessitated holding the whole cell patch on each cell for an additional 20-30 minutes. Such an extension would have significantly limited the time available for exploring both the spiking and membrane voltage responses over the large parameter space used in this study. Furthermore, given much previous work showing that RGC responses to HFS are not sensitive to synaptic blockers [Ref 1-4], we chose to focus on pursuing new results rather than re-testing the effects of synaptic isolation.

[Ref 1] Twyford, P., Cai, C. & Fried, S. Differential responses to high-frequency electrical stimulation in ON and OFF retinal ganglion cells. J Neural Eng 11, 025001 (2014).

[Ref 2] Muralidharan, M. et al. Neural activity of functionally different retinal ganglion cells can be robustly modulated by high-rate electrical pulse trains. J Neural Eng 17, 045013 (2020).

[Ref 3] Freeman, D.K., Eddington, D.K., Rizzo, J.F., 3rd & Fried, S.I. Selective activation of neuronal targets with sinusoidal electric stimulation. J Neurophysiol 104, 2778-2791 (2010).

[Ref 4] Cai, C., Twyford, P. & Fried, S. The response of retinal neurons to high-frequency stimulation. J Neural Eng 10, 036009 (2013).

Line 423:

It is important to note that the contribution of synaptic input to HFS-induced suppression (or other mechanisms besides depolarization block) may be more significant in other neuronal systems. For example, in the neocortex and hippocampus, synaptic transmission plays a critical role in paroxysmal depolarization and depolarization block of the cells, and depending on cell type (e.g., excitatory vs. inhibitory neurons) or size, the neurons have different depolarization block properties mediated by synaptic input [Ref 5]. Given their high depolarization block sensitivity to synaptic input, the HFS-induced suppression in these neurons might result from synaptic transmission triggered by HFS, with or without a contribution from the intrinsic ion channel dynamics of the cell membrane.

[Ref 5] Tryba, A.K. et al. Role of paroxysmal depolarization in focal seizure activity. J. Neurophysiol. 122, 1861-1873 (2019).

2) [#4] *The readability is improved but addition of essential graphs in the supplement (e.g., Suppl. Fig. 3), rather than include them as a panel in an existing figure, doesn't necessarily help in my opinion.*

The supplementary figure 3 in the previous version is now added to the main figure (Figure 1E and F, and Figure 3E and F).

3) *The above comment is also relevant for Suppl. Figs. 5 and 6.*

The supplementary figure 5 and 6 in the previous version is now added to the main figure 2D and 11, respectively.

4) [#11] *The model description is improved. I do have an additional reaction; See #11, Rev#3 below.*

Thanks for the feedback. Our response to #11 from Rev#3 is below.

5) [#14] *I do not understand the reply and how this was addressed in the revision.*

To be effective, artificial stimulation must depolarize the neuronal membrane strongly enough to activate voltage-gated sodium channels. If enough channels are activated, the regenerative mechanism described by Hodgkin and Huxley is activated and an action potential is generated. To generate multiple spikes however, it is necessary for membrane voltage to repolarize, i.e., so that inactivated sodium channels have a chance to de-inactivate. If all other variables are held constant, conditions that enable effective repolarization will lead to the highest levels of spiking.

We believe that stimulus conditions that lead to hyperpolarization (such as those shown in Figure 3) are more effective because such levels of V_m are more effective in de-inactivating voltage-gated sodium channels and thus the supply of channels available to generate spiking remains robust. The onset of each stimulus still depolarizes the cell enough to trigger a spike while the cumulative effect of prolonged (hyperpolarizing) potassium currents following the relatively short duration of (depolarizing) sodium currents leads to hyperpolarization (and stronger dis-inhibition of sodium channels). We did not perform physiological measurements to unravel the exact mechanisms and we therefore convey to the reader that our comments here require additional validation. In the revised manuscript, we have included a more detailed description of the increased firing associated with hyperpolarized levels of V_m . We also added the details here to the Discussion along with some comments about the potential value of further investigation into this finding.

Line 434:

Responses to low-frequency stimulation

In addition to our primary focus on HFS responses, we also made several intriguing observations from the responses to low-frequency stimulation (LFS; < 1000 Hz). LFS that produced stronger hyperpolarization of V_m also produced a larger number of spikes (Figures 1D and 3D), i.e., stronger spiking responses arose from hyperpolarized V_m (not depolarized). We did not directly investigate the reason(s) for this, but speculate that hyperpolarization of V_m produces a larger number of available (de-inactivated) voltage-gated sodium channels, which in turn better ensures the generation of spiking during each cathodal phase of sinusoidal stimulation. Another interesting observation was that ON and OFF α sustained RGCs show different sensitivities not only to HFS, but also to lower frequencies. For example, in response to 500 Hz, the peak non-monotonic responses occurred at weaker stimulus amplitudes in ON cells vs. OFF cells (compare the blue and red asterisks in Figure 11A). Furthermore, at the transition point where non-monotonic responses shifted to monotonic responses (indicated by the filled circles in Figure 11A), spike counts were considerably smaller in ON cells compared to OFF cells (10.7 ± 2.6 spike/s vs. 62.9 ± 20.43 spike/s). The two cell types also showed different depolarization sensitivity; weak stimulus amplitudes induced higher depolarization in ON cells compared to OFF cells (Figure 11B). While the small sample sizes for both cell types in Figure 10B and C impose limitations on conducting statistical analysis, these results nevertheless suggest that LFS may be potentially useful for selective activation.

6) [#6, Rev#3] It is not clear to me how the reply to point 2 is reflected in the revised text.

In the revised manuscript, we have explicitly stated that our study does not enable unequivocal preferential activation, nor does it allow absolute selectivity among all RGC types. Instead, our findings offer new insights into the biophysical mechanism underlying the complex and unique responses elicited by HFS, results that may lead to enhanced selectivity.

Line 398:

It is important to emphasize that our results do not demonstrate unequivocal preferential activation, nor do they suggest that HFS enables selectivity of all RGC types. However, the novel insights into the biophysics underlying complex and unique responses to HFS presented here provide an improved foundation for improving stimulation strategies with retinal prostheses, as well as for other neural engineering applications.

7) [#11, Rev#3] Bringing up different types of currents can be relevant for the physiological interpretation. However, the modeling results are to be interpreted in light of the limitations of Hodgkin & Huxley type models. Because not all currents are known, certainly not in sufficient detail, parameters must be fudged a bit (in physics one might call this renormalization ... :-). I miss an honest assessment of that limitation.

We agree with the reviewer's comment. Indeed, a more realistic neuronal model, accounting for comprehensive details about the cell's anatomy and ion channel properties, would likely add additional understanding of the responses to HFS and the different sensitivities across cell types. However, as the reviewer noted, the challenges in capturing sufficient detail to significantly improve computational models necessitate a cautious interpretation of the results. We have tried to better acknowledge this limitation in the revised manuscript.

Line 339:

We cannot rule out the possibility that other factors contribute as well, e.g., multiple subtypes of voltage-gated potassium (and other) voltage sensitive channels can be expressed at multiple locations within individual neurons, and these may interact in complex ways that are difficult to replicate in a single-compartment model. Thus, while our modeling results suggest that intrinsic morphological and biophysical features do indeed shape the responses to HFS, further studies with more realistic neuronal models are needed before making more definitive conclusions about the mechanism(s).

Line 354:

It will be important to directly assess efficacy of HFS in the degenerate retina given that the single compartment model used here does not encompass the full spectrum of synaptic and intrinsic properties that change during retinal degeneration.

Reviewer #2

1) It is interesting that there is a domain in which on-RGCs are suppressed while off-RGCs respond under HFS which leads to selective targeting of cells. Since this sensitivity difference is mentioned in the title/abstract and is one of the authors main focuses in this paper, it is worth to show computationally the sensitivity differences across two RGC types under HFS by modelling.

What factor determines the sensitivity differences in response to HFS in these two types of the cells from computational study?

We agree that this is an important and relevant question and ran some additional simulations to provide such insights. We want to point out first however, that there are several challenges associated with exploring the underlying mechanism(s) using a computational model. For example, if we speculate that the sensitivity differences arise from differences in the distribution of voltage-gated potassium channels, we are faced with questions about the specifics of such differences, e.g., location within the neuron as well as the specific type of potassium channel. Even if we assume that all relevant changes are within the axon initial segment (AIS), we would need to systematically investigate sensitivity to the length, specific location within the AIS (e.g., distance from the soma), and the channel density, i.e., how much each property shapes sensitivity. From there, we would need to consider that there are thought to be several different types of voltage-gated potassium channels within the AIS and possibly other types in other parts of the axon and soma. Multiple types of sodium and calcium channels could also be contributing. Many details on expression level and channel function are still not known and so a considerable amount of background investigation would be needed, in both ON and OFF types, before we could develop an anatomically realistic model.

Because this is an important point, we used a somewhat modified approach to gain some insights. Specifically, we continued to use the single compartment model and explored how changes to model parameters influenced sensitivity to HFS (see Figures 10A & B in the revised manuscript). We found that increasing the density of voltage-gated potassium channels in the model increased the stimulus amplitude at which peak responses occurred, raising the possibility that OFF RGCs have higher potassium currents than ON RGCs. We also observed that enlarging the soma size led to peak responses at higher current amplitudes in our model. These findings suggest that both morphological features and channel properties could significantly influence HFS responses, potentially accounting for the differences observed between ON and OFF cells, and providing a roadmap for further investigation.

We have incorporated these new modeling findings and their explanations into the manuscript.

Line 332:

To evaluate whether sensitivity differences between ON and OFF cells would emerge from changes to the intrinsic properties within the single compartment model, we ran an additional series of simulations in which the size of the soma was varied while all other parameters were held constant (Figure 10A); in an analogous set of experiments, we then varied the conductance of potassium channels with all other parameters constant (Figure 10B). In both cases, the amplitude level at which depolarization block occurred was sensitive to changes in the individual parameters. These results therefore support the notion that intrinsic differences between types can mediate sensitivity differences to HFS for different cell types although we cannot rule out the possibility that other factors contribute as well, e.g., multiple sub-types of voltage-gated potassium (and other) voltage sensitive channels can be expressed at multiple locations within individual neurons, and these may interact in complex ways that are difficult to

replicate in a single-compartment model. Thus, while our modeling results suggest that intrinsic morphological and biophysical features do indeed shape the responses to HFS, further studies with more realistic neuronal models are needed before making more definitive conclusions about the mechanism(s).

Figure 10. Variation in non-monotonic response curves depending on modeling parameters.

The spiking responses to 2000 Hz stimulation were simulated using a single compartment model. While maintaining other parameters consistent with those in Figure 6, we investigated the impact of changes in cell size (A), potassium channel density (B), and noise level (C) on the response curves. Blue traces indicate standard parameters.

2) What do authors expect when HFS is applied to different neurons in different regions of the brain with burst spiking, high spontaneous firings (like Purkinje cells)? It is good to show the dependency of the effect on different regimes of spontaneous firing by computational modelling.

This is an intriguing and relevant question that we had not considered. Several forms of retinal degeneration have been shown to increase spontaneous firing rates in RGCs; because these diseases lead to blindness, the prosthesis may need to remain effective across different levels of baseline firing. We agree that such insights might also be of interest to neural prostheses in other parts of the CNS.

The heightened spontaneous firing observed in Purkinje cells and RGCs in degenerated retinas can be attributed to various biological factors, including intrinsic properties and synaptic inputs. However, our simplified single-compartment model does not capture the unique characteristics of these cells, which limits its ability to yield meaningful insights into such cases. Despite this limitation, we attempted to simulate an increase in the spontaneous firing rate of our model cell by enhancing the noise level, which is intended to reflect stronger baseline synaptic input. Interestingly, this adjustment did not have much of an effect on the spiking produced by HFS (Figure 12C). We point out to the reader that our use of a single compartment model does not capture all of the underlying biology and biophysical properties; thus, even though these results suggest HFS remains robust with increased baseline firing, further investigation will be required before a definitive conclusion can be reached. We also remind the reader that this finding is consistent with earlier work from Margolis (2008) that used physiological measurements to show that the intrinsic, spike-generating mechanism(s) in these specific types of RGCs is not altered even at the highest levels of increased baseline firing that arise in the degenerate retina.

We have included these modeling results and the relevant discussion in our revised manuscript.

Line 345:

Does baseline firing influence sensitivity to HFS?

In the retina, conditions like retinitis pigmentosa and age-related macular degeneration alter the synaptic inputs to RGCs, leading to increases in the rate of spontaneous firing [Ref 1 and 2]. Because these diseases lead to blindness, and thus may require a retinal prosthesis, we questioned whether response sensitivity to HFS is altered by changes in baseline firing levels by performing one additional set of simulations. We modified the noise level parameters in our model to increase the spontaneous firing rates but found only minor changes in the responses to HFS (Figure 10C). These results are consistent with a previous study in RGCs that showed that the spike generation mechanism remains largely intact in the presence of the increased background spiking that occurs during retinal degeneration [Ref 3-5]. While these results therefore suggest that the sensitivity to HFS may remain consistent in the diseased retina, it will be important to directly assess efficacy of HFS in the degenerate retina given that the single compartment model used here does not encompass the full spectrum of synaptic and intrinsic properties that change during retinal degeneration.

[Ref 1] Stasheff, S.F., Shankar, M. & Andrews, M.P. Developmental time course distinguishes changes in spontaneous and light-evoked retinal ganglion cell activity in rd1 and rd10 mice. J. Neurophysiol. 105, 3002-3009 (2011).

[Ref 2] Goo, Y.S., Park, D.J., Ahn, J.R. & Senok, S.S. Spontaneous oscillatory rhythms in the degenerating mouse retina modulate retinal ganglion cell responses to electrical stimulation. Frontiers in cellular neuroscience 9, 512 (2016).

[Ref 3] Sekirnjak, C. et al. Loss of responses to visual but not electrical stimulation in ganglion cells of rats with severe photoreceptor degeneration. J. Neurophysiol. 102, 3260-3269 (2009).

[Ref 4] Yoon, Y.J. et al. Retinal degeneration reduces consistency of network-mediated responses arising in ganglion cells to electric stimulation. IEEE Trans. Neural Syst. Rehabil. Eng. 28, 1921-1930 (2020).

[Ref 5] Margolis, D.J., Newkirk, G., Euler, T. & Detwiler, P.B. Functional stability of retinal ganglion cells after degeneration-induced changes in synaptic input. J. Neurosci. 28, 6526-6536 (2008).

3) The authors examined 1s duration for HFS. It is good to discuss how the result (inhibition of neural activity) changes for different duration of stimulation (lower/ higher than 1s)?

We agree that readers may want to understand if preferential activation of neurons can be achieved within a shorter stimulus duration. Previous studies, both inside and outside the retina, have shown that the effect of HFS is rather quick, particularly when compared to slower-acting chemical drugs. To explore sensitivity to shorter duration stimuli, we reanalyzed our recordings, focusing on spikes generated within the first 100 ms post-stimulus onset, as opposed to considering the entire stimulus duration of 1 second. We observed that at high HFS amplitudes, cells typically entered a depolarization block within the first 100 ms post-stimulation onset (Figure 4C). As a result, the pattern in the ON and OFF response curves to shorter duration stimuli (e.g., 100-ms duration, Figure 10A) was very similar to that for the full 1-second duration (Figure 5A). We have included this additional analysis in our manuscript, along with relevant explanations.

Line 258:

These results were not dependent on use of a 1-second duration over which spikes were counted since cells typically entered depolarization block after only a few spikes following the onset of stimulation (Figure 4C) and therefore ON vs. OFF response curves for the first 100 ms post-stimulus onset (Figure 5G) were similar to those for 1 second (Figure 5A).

Figure 5G. For the same dataset used in Figure 5A, the number of spikes elicited only during the first 100 ms post-stimulus onset were counted. Error bars indicate the standard error of the mean (SEM).

4) All the experimental results are taken from explanted mice (which would be better to mention explicitly in the caption of relevant figures). The result can be different for in vivo cells in living animals which need to be discussed.

In the captions of figures displaying physiological data (Figures 1-5, 10, and 11), we have clearly indicated that the results were obtained from explanted mouse retinas and/or in-vitro setup. Additionally, we have stated the constrained scope of our study as follows:

Line 431:

It is also important to note that our physiological results were exclusively obtained from explanted mouse retinas, and thus other mechanisms besides depolarization block could also be relevant in in-vivo conditions or other neuronal systems.

5) *It is better for the reader to know about the error bars (whether standard deviation or SEM) in each figure caption (Figs. 1, 3, 4,...)*

Clarifications regarding the error bars have been added to the captions of Figures 1, 3, 4, and 10.

6) *Figure 6B: Is the frequency for the last graph 1500 Hz or 2000 Hz? (Text in caption and figure are not consistent.)*

The typo in the caption has been corrected.

7) *Line 257-258: “monotonic responses for low frequencies of stimulation (Figure 6B, top), non-monotonic responses for high frequencies (Figure 6B, 257 bottom) and combination of both for intermediate frequencies (Figure 6B, middle).”*

There is no top/middle/bottom in Figure 6B.

Thank you for highlighting the error in our figure arrangement. We have now accurately corrected the terms from top/middle/bottom to left/middle/right.

8) *“To better understand this apparent discrepancy, we compared the level of depolarization that arose in ON vs. OFF cells as a function of the amplitude of HFS and found that a given increase in amplitude produced a larger depolarization in OFF (vs. ON) cells (Figure 5F).”*

A given increase in amplitude produces a larger depolarization in ON cells (vs. OFF).

Thank you for pointing out the error. We have revised the text to accurately state ‘a larger depolarization in ON (vs. OFF) cells’.

REVIEWERS' COMMENTS:

Reviewer #1 (Remarks to the Author):

The authors addressed the critiques.

Reviewer #2 (Remarks to the Author):

The paper now is well written and interesting to read.

A few errors/typos that I noticed:

The order of supplementary figures 2 and 3, need to be changed.

Line 171: Supplementary figure 2-> 3

Line 207: Supplementary figure 3 -> 2

Line 840: Pearson

Reviewer's comments:

Reviewer #1

The authors addressed the critiques.

We appreciate the reviewer's comments throughout the review process. Thanks to your feedback, we have been able to enhance the manuscript, particularly in improving clarity and access to a wider readership.

Reviewer #2

The paper now is well written and interesting to read.

A few errors/typos that I noticed:

1) The order of supplementary figures 2 and 3, needs to be changed.

Line 171: Supplementary figure 2-> 3

Line 207: Supplementary figure 3 -> 2

Thank you for bringing the error to our attention. We've revised the manuscript to ensure that the text aligns with the figure numbering.

2) Line 840: Pearson

The typo has been corrected.